# Emerging Role of miR-21-5p in Neuron–Glia Dysregulation and Exosome Transfer Using Multiple Models of Alzheimer’s Disease

**DOI:** 10.3390/cells11213377

**Published:** 2022-10-26

**Authors:** Gonçalo Garcia, Sara Pinto, Sofia Ferreira, Daniela Lopes, Maria João Serrador, Adelaide Fernandes, Ana Rita Vaz, Alexandre de Mendonça, Frank Edenhofer, Tarja Malm, Jari Koistinaho, Dora Brites

**Affiliations:** 1Neuroinflammation, Signaling and Neuroregeneration Lab, Research Institute for Medicines (iMed.ULisboa), Faculty of Pharmacy, Universidade de Lisboa, 1649-003 Lisbon, Portugal; 2Department of Pharmaceutical Sciences and Medicines, Faculty of Pharmacy, Universidade de Lisboa, 1649-003 Lisbon, Portugal; 3Instituto de Medicina Molecular, Faculdade de Medicina, Universidade de Lisboa, 1649-028 Lisbon, Portugal; 4Central Nervous System, Blood and Peripheral Inflammation Lab, Research Institute for Medicines (iMed.ULisboa), Faculty of Pharmacy, Universidade de Lisboa, 1649-003 Lisbon, Portugal; 5Faculty of Medicine, Universidade de Lisboa, 1649-028 Lisbon, Portugal; 6Department of Genomics, Stem Cell Biology and Regenerative Medicine, Center for Molecular Biosciences, University of Innsbruck, 6020 Innsbruck, Austria; 7A.I. Virtanen Institute for Molecular Sciences, University of Eastern Finland, 70211 Kuopio, Finland; 8Neuroscience Center, Helsinki Institute of Life Science (HiLIFE), University of Helsinki, 00014 Helsinki, Finland

**Keywords:** CSF miRNAs, exosomal miRNAs, glial activation, hippocampal neuroblastoma transplantation, immunostimulated astrocytes, inflammation-associated miRNAs, inflammatory mediators, iPSC-derived AD models, *PSEN1ΔE9* expressing cells, SH-SY5Y *APP SWE* cells

## Abstract

Alzheimer’s disease (AD) is a neurodegenerative disorder associated with neuron–glia dysfunction and dysregulated miRNAs. We previously reported upregulated miR-124/miR-21 in AD neurons and their exosomes. However, their glial distribution, phenotypic alterations and exosomal spread are scarcely documented. Here, we show glial cell activation and miR-21 overexpression in mouse organotypic hippocampal slices transplanted with SH-SY5Y cells expressing the human *APP695 Swedish* mutation. The upregulation of miR-21 only in the CSF from a small series of mild cognitive impairment (MCI) AD patients, but not in non-AD MCI individuals, supports its discriminatory potential. Microglia, neurons, and astrocytes differentiated from the same induced pluripotent stem cells from *PSEN1ΔE9* AD patients all showed miR-21 elevation. In AD neurons, miR-124/miR-21 overexpression was recapitulated in their exosomes. In AD microglia, the upregulation of iNOS and miR-21/miR-146a supports their activation. AD astrocytes manifested a restrained inflammatory profile, with high miR-21 but low miR-155 and depleted exosomal miRNAs. Their immunostimulation with C1q + IL-1α + TNF-α induced morphological alterations and increased S100B, inflammatory transcripts, sAPPβ, cytokine release and exosomal miR-21. PPARα, a target of miR-21, was found to be repressed in all models, except in neurons, likely due to concomitant miR-125b elevation. The data from these AD models highlight miR-21 as a promising biomarker and a disease-modifying target to be further explored.

## 1. Introduction

Accompanied by the increase in life expectancy, Alzheimer’s disease (AD) has become a major global public health problem with an increasing burden and massive socioeconomic and political impacts [1]. Progress in target identification and drug discovery, biomarker evidence and patient stratification to reduce heterogeneity may increase the success of new therapies in clinical trials for AD [2,3]. To better follow patients with early neurocognitive complaints and their eventual progression to AD, the concept of mild cognitive impairment (MCI) based on AD biomarkers was considered clinically useful and shown to impact patient management [4,5]. At present, AD diagnosis is based on the existence of dementia, brain volume changes detected by magnetic resonance imaging (MRI), and brain amyloid β-protein (Aβ) or Tau deposition detected by positron emission tomography (PET), together with an Aβ42 decrease and a phosphorylated/total Tau increase in the cerebrospinal fluid (CSF) [6]. New analytical tools, which include markers of synaptic dysfunction and inflammation in the blood, as well as microRNAs (miRNAs), suggest that AD may be diagnosed with accuracy in individual patients [7].

Dementia in AD progresses with neurodegeneration, but the underlying mechanisms of neuronal dysfunction and death remain poorly understood [8]. Moreover, current data supports the idea that AD pathology also involves dysregulated crosstalk between neurons, astrocytes and microglia that is extended to endothelial cells and oligodendrocytes [9]. With many of these neuropathological events taking place in the hippocampus and cortex, the loss of cognitive functions is an inevitable hallmark of AD patients. Astrocytes are known to contribute to neuroinflammation, oxidative stress and calcium imbalance in AD [10]. The aberrant accumulation of hypertrophic astrocytes was found around Aβ plaques [11], while Aβ42 was shown to impair the function of the excitatory amino acid transporter (GLT-1) and the glutamate/aspartate transporter (GLAST) in hippocampal astrocytes [12]. Despite their susceptibility to Aβ, astrocytes play an important role in its clearance via extracellular protease degradation [13]. Likewise, microglia, the brain-resident immunocompetent macrophages, also recognize Aβ oligomers via receptors such as CD36, toll-like receptor 4 (TLR4) and TLR6 and triggering receptor expressed on myeloid cells 2 (TREM2), acquiring specific AD-associated phenotypes [14,15,16]. As one of the most relevant consequences, the loss of microglial phagocytic capacity leads to further Aβ accumulation and aggregation, accelerating the progression of AD [17]. Together, activated microglia and reactive astrocytes accumulate in the surrounding area of senile plaques, contributing to the severity of the disease by dysregulating the inflammatory response [18].

Several experimental models of AD have been developed and intensively explored in search for novel pathological features and more effective therapies and targets, with the objective of better translating the results of preclinical studies to clinical reality [19,20]. From classic immortalized cell lines and post-mortem brain tissue to more recent animal models and human brain organoids, all these models contributed to our current comprehension of AD pathogenesis, though there is still much to be understood. The generation of induced pluripotent stem cells (iPSCs) constitutes a major scientific breakthrough and offers numerous new ways to study AD and other diseases [21,22,23]. Their potential to differentiate into any type of cell and the possibility to ultimately generate organ-like structures have been extensively explored and optimized over the last decade [24]. In AD, once generated from familial or sporadic patients, iPSC-based models have been used to explore and stratify patient-specific phenotypes using different cell types, including neurons [25], microglia [26] and astrocytes [27,28].

In iPSC-derived neurons of frontotemporal dementia (FTD) and amyotrophic lateral sclerosis (ALS) patients, miRNA(miR)-9 downregulation was found [29], and our own data revealed the overexpression of miR-124, miR-125b and miR-21 in iPSC-derived neurons carrying the *PSEN1ΔE9* deletion (iNEU-PSEN) [25]. Interestingly, the same miRNA dysregulation was found in SH-SY5Y cells (SH) expressing the human Amyloid Precursor Protein (APP)695 *Swedish* mutation (SWE). Among these, miR-125b is associated with Tau phosphorylation [30], miR-124 with neuronal function and APP processing [25,31], and miR-21 to the regulation of toxicity mediated by Aβ oligomers, both in vitro and in vivo [32]. Despite its nonspecificity [33], miR-21 is considered a versatile regulator in the progression of CNS disorders [34] and was suggested to have both harmful and beneficial effects [35,36,37]. In general, miRNAs can be released by cells as free species directly into their secretome or selectively enriched in small extracellular vesicles (exosomes, diameter size <150 nm) that, when internalized by target cells, affect their biological functions [38,39,40,41]. Particularly, miR-124 and miR-21 were found to be elevated in exosomes from SWE cells, and miR-124 was upregulated in those from iNEU-PSEN cells [25]. However, there is still a long way to go before miR-21 and miR-124 are considered as disease-modifying targets in the AD context, as miRNAs’ role as biomarkers and targets is still a matter of discussion, and little is known about their regulation and dynamics in the field.

To explore the specific contribution of individual cell types and immune-related miRNAs to inflammatory responses in AD [42,43], we took advantage of having iPSC-derived neurons, astrocytes and microglia generated from patients with the *PSEN1ΔE9* mutation and controls; a coculture system of SH or SWE cells transplanted into mouse hippocampal slices; and the cerebrospinal fluid (CSF) of a series of patients with MCI due to AD (MCI-AD) and non-AD MCI patients (MCI-Ctrl). A specific emphasis was given to the representation of miR-21 among the immune-selected miRNAs in the tested conditions and to its trafficking in exosomes. Unique miR-21 upregulation, together with microglial activation and astrocyte reactivity, was identified in 3D organotypic hippocampal cultures transplanted with SWE cells. Likewise, miR-21 was found to be upregulated in the CSF of MCI-AD patients, but not in that of MCI-Ctrl individuals, as well as in all iPSC-derived cell types. When exosomes derived from AD neurons and astrocytes were evaluated for their miRNA content, only those from neurons were enriched in miR-21, while sensitization with a cocktail composed by complement component C1q + interleuklin-1 alpha + tumor necrosis factor alpha (C1q + IL-1α + TNF-α), known to induce neurotoxic reactive astrocytosis [44], was required to produce the same result in exosomes from AD astrocytes. Our findings highlight the upregulation of miR-21 in different AD models, suggesting its potential as a new therapeutic target.

## 2. Materials and Methods

### 2.1. Animals and Ethics

Wild-type (*WT*) B6SJLF1/J mice were purchased from The Jackson Laboratory (Bar Harbor, ME, USA). Maintenance and handling took place at the Instituto de Medicina Molecular João Lobo Antunes (IMM) animal house facilities of the Faculty of Medicine, University of Lisbon, Portugal, where the colony was established. All animals were maintained on a 12 h light/12 h dark cycle and received food and water ad libitum. The average number of animals per cage was 4 to 5. The present study was performed in accordance with the European Community guidelines (Directives 86/609/EU and 2010/63/EU, Recommendation 2007/526/CE, and European Convention for the Protection of Vertebrate Animals used for Experimental or Other Scientific Purposes ETS 123) and Portuguese Laws on Animal Care (Decreto-Lei 129/92, Portaria 1005/92, Portaria 466/95, Decreto-Lei 197/96 and Portaria 1131/97). The protocols used in this study were approved by the Portuguese National Authority (General Direction of Veterinary) and the Ethics Committee of the IMM. According to the 3R principle, every effort was made to minimize the number of animals used and their suffering.

### 2.2. Organotypic Hippocampal Cultures

The preparation of organotypic hippocampal slice cultures was performed following our previous publication with minor modifications [45]. Briefly, *WT* B6SJLF1/J mouse pups (6–7 days) were sacrificed, their brains were removed under sterile conditions, and the two hippocampi were isolated and cut into 400 µm coronal sections using a McIlwain Tissue Chopper^®^ (Gomshall, Surrey, UK). A set of 4 slices were placed in the upper face of the insert membrane (BD Falcon, Lincoln Park, NJ, USA). The slices were then transferred onto polycarbonate membranes of 0.4 mm in the upper chamber of a Transwell^®^ tissue insert (Becton Dickinson Falcon, Lincoln Park, NJ, USA) and further placed into a six-well plate (Falcon model 3502, Becton Dickinson). Next, slices were cultured in 1.5 mL of culture medium consisting of 50% MEM, 24% heat-inactivated horse serum, 24% EARL’s, 1% AB/AM and 1% L-glutamine, supplemented with 0.02 mg/mL insulin and 0.016 mg/mL ascorbic acid (all from Sigma-Aldrich, St. Louis, MO, USA), and maintained at 37 °C in 5% CO_2_ culture conditions. On each of the following 3 days, half of the medium was discarded and replaced by fresh medium. All experiments with hippocampal slices started after 4 days in vitro (DIV) to allow for trauma cut recovery.

### 2.3. Human Neuroblastoma Cell Culture

Human neuroblastoma SH and SWE cells were a gift from Professor Anthony Turner. Cells were routinely tested for mycoplasma contamination and cultured in Dulbecco’s Modified Eagle’s Medium (DMEM) supplemented with 10% fetal bovine serum (FBS) and 2% AB/AM (Thermo Fisher Scientific, Waltham, MA, USA) in T75 flasks. Then, cells were maintained at 37 °C in a humidified atmosphere of 5% CO_2_, as routinely conducted in our laboratory [24,40].

### 2.4. Human Neuroblastoma Live-Cell Staining and Transplantation

SH and SWE cells were chemically detached using 0.25% trypsin and counted. Then, a cell suspension was prepared with the final concentration of 1 × 10^6^ cells/mL and incubated with 5 µM CellTracker™ Red CMTPX Dye (Cell tracker, Themo Fisher Scientific) for 30 min at 37 °C in a humidified atmosphere of 5% CO_2_. Then, the cells were spun down, the excess dye was discarded, and the cells were resuspended in the same volume of fresh media to maintain an equal working concentration (1 × 10^6^ cells/mL). Retinoic acid was used for neuronal differentiation as previously indicated [25]. Lastly, a total of 1 × 10^4^ neuroblastoma cells were seeded, in small volumes of 10 µL, on the top of 4 DIV hippocampal slice cultures, as previously described [46]. The mixed neuroblastoma–hippocampal slice culture was maintained for an additional 4 DIV, and the cell tracker dye was confirmed to be stable during this period.

### 2.5. MCI Patients, Clinical Assessment and CSF Collection

The dataset and samples used in this study were obtained during the development of the project PTDC/MED-NEU/27946/2017 from Fundação para a Ciência e Tecnologia. This project was approved by the local ethics committee, and the participants provided their written informed consent. Participants were recruited at Memoclínica, a private memory clinic in Lisbon, and submitted to the mini-mental state examination [47] and to the Clinical Dementia Rating (CDR) [48]. Patients fulfilled the criteria adopted for MCI due to AD—high likelihood, corresponding to the highest level of certainty (National Institute on Aging—Alzheimer’s Association workgroups [5]) (Table 1). A high likelihood requires meeting clinical and cognitive criteria; biomarkers of Aβ deposition (low CSF Aβ42 levels or positive for brain amyloid on Pittsburgh B Compound PET scan); and biomarkers of neuronal injury (at least two positive markers: high CSF total Tau or hyperphosphorylated Tau, medial temporal lobe atrophy detected by volumetric measures or visual rating, or temporoparietal hypometabolism detected by fluorodeoxyglucose PET imaging). Controls were subjects meeting clinical and cognitive criteria for MCI but showing none of the biomarkers of either Aβ deposition or neuronal injury indicated in Table 1. Lumbar puncture [49] and CSF handling [50] were performed following the established standard operating procedures. Briefly, after their collection, the samples were centrifuged at 2000× *g* for 10 min at room temperature (RT) to pellet any cells or debris. Following centrifugation, CSF aliquots of 500 μL were stored in code-labeled polypropylene tubes at −80 °C until analysis.

### 2.6. RNA Isolation and Purification from CSF

CSF aliquots from MCI patients were used for RNA isolation and purification using the total RNA Purification kit (Norgen, Thorold, CA, USA) following the manufacturer’s specific recommendations for CSF. The concentration of RNA was determined using a Nanodrop^®^ ND-100 Spectrophotometer (NanoDrop Technologies, Wilmington, DE, USA), and high-quality RNA samples were stored at −80 °C until processing.

### 2.7. Culture and Differentiation of Patient iPSCs

Human iPSCs and astrospheres were generated from AD male/female patients (carrying the *Finnish PSEN1ΔE9* mutation, AD2 and AD3) and healthy female controls (Ctrl1 and Ctrl3), all presenting the *ε3/ε3* genotype (Table 2), at Jari Koistinaho’s laboratory (University of Helsinki and University of Eastern Finland). The same iPSC samples were additionally obtained from Frank Edenhofer’s laboratory (University of Innsbruck), and microglia progenitors (AD3 and Ctrl3) were obtained from Tarja Malm’s laboratory (University of Eastern Finland), thanks to the JPco-fuND 2015 project MADGIC and covered by a Material Transfer Agreement contract. Informed consent was obtained from all subjects before sample collection. iPSC lines were generated after the approval of the committee on Research Ethics of Northern Savo Hospital District (123/2016). Demographic information of each iPSC line is summarized in Table 2. Characterization for genetic stability and pluripotency markers of these iPSCs was previously documented in studies using iPSC differentiation into astrocytes and microglia for the screening of AD-associated glial aberrancies [26,27]. Briefly, iPSCs were grown on Matrigel-coated plates (Corning, Corning, NY, USA) in Essential 8 (E8) medium and passaged with 0.5 mM EDTA. Freshly passaged cells were cultured with 5 mM Y-27632 ROCK inhibitor (Selleckchem, Houston, TX, USA). Different protocols, as will be described below, were applied to differentiate, and maturate astrocytes, neurons and microglia from iPSCs, but no differences were found between patient cells and matched controls when assessed for the gene expression of Ki67, whose protein is widely used as a proliferation marker (data not shown).

### 2.8. Primitive Hematopoietic Differentiation of iPSCs into Microglia

For the differentiation of Ctrl and AD iPSCs into microglia-like cells (MG, microglia), a recently established protocol based on primitive hematopoiesis induction was used [26]. Early mesodermal differentiation and the expansion of hematopoietic precursor cells (HPCs) were carried out at Tarja Malm’s laboratory. For early mesodermal differentiation, on day 0 (D0), confluent iPSCs were washed, detached using EDTA, counted, split into Matrigel-coated plates, and cultured under low-oxygen conditions (5% O_2_/5% CO_2_) until D2 (see Appendix A), the time when the mesodermal marker brachyury was identified [26]. Hematovascular mesodermal differentiation was processed from D2 until D4, when cells returned to normoxia conditions; on D3, dead cells were removed, and on D4, the medium was changed to the erythro-myeloid progenitor (EMP) medium (DF3S supplemented with 50 ng/mL FGF2, 5 µg/mL insulin, 50 ng/mL VEGF, 50 ng/mL TPO, 10 ng/mL SCF, 50 ng/mL IL-6 and 10 ng/mL IL-3 (all from Peprotech, Rocky Hill, NJ, USA), and the cells were transferred into a humidified normoxic incubator (5% CO_2_, 37 °C). On D7, EMPs started to form on top of the monolayer in the dish, and on D8, “blooming” EMPs (see Appendix A) were gently detached using a pipette, frozen in cryovials at −80 °C and sent to our laboratory in Lisbon, Portugal.

After thawing, we expanded the EMP/microglial progenitor culture until D10. For this purpose, EMPs were routinely cultured in 10 cm ultra-low-attachment (ULA) dishes (Corning) in the presence of microglial progenitor medium (Iscove’s Modified Dulbecco’s Medium, IMDM) (Thermo Fisher Scientific) supplemented with 10% FBS, 1% P/S, 5 µg/mL Insulin, 100 ng/mL IL-34 and 5 ng/mL M-CSF) in a humidified atmosphere containing 5% CO_2_ at 37 °C. Then, the medium was switched to primitive macrophage (PM) medium (IMDM supplemented with 10% FBS, 0.5% P/S, 10 ng/mL IL-34 and 10 ng/mL M-CSF). Every other day, the total medium volume was discarded and replaced with fresh PM medium. During this time, the cells started to attach to the bottom of ULA plates as they matured. From D16–D23, primitive microglia were chemically detached from the ULA plates with StemPro^®^ Accutase (Thermo Fisher Scientific) for 5 min at 37 °C and seeded into 12-well dishes coated with Poly-D-Lysine at a final concentration of 70,000 cells per well in PM medium. Half of the medium was changed daily until the experiments were concluded. Cells on D24 exhibited phagocytic ability and several positive markers, such as transmembrane protein 119 (TMEM119), ionized calcium-binding adapter molecule 1 (Iba1), CX3C motif chemokine receptor 1 (CX3CR1), milk fat globule-EGF factor 8 protein (MFGE8), Cd11b, triggering receptor expressed on myeloid cells 2 (TREM2) and Arginase-1 (Arg-1), as documented in the Results section. Compared to iPSC-derived neurons and astrocytes (either sensitized or not), the TREM2 gene was preferentially expressed in microglia (see Appendix A). In addition, the expression of synaptophysin (SYP) and glial fibrillary acidic protein (GFAP) genes was negligible, and that of connexin 43 (Cx43) scarcely identified (Appendix A).

### 2.9. Neural Induction of iPSCs and Sphere Formation

The neural differentiation protocol was modified from previously described dual SMAD inhibition protocols [51,52], as published by us [25]. Differentiation was started by changing to neural differentiation medium (NDM) consisting of DMEM/F12 and Neurobasal (1:1), 1% B27 without vitamin A, 0.5% N2, 1% Glutamax and 0.5% penicillin/streptomycin (50 IU/50 mg/mL) (all from Invitrogen, Waltham, MA, EUA), supplemented with dual SMAD inhibitors: 10 mM SB431542 (Sigma-Aldrich) and 200 nM LDN193189 (Selleckchem). The medium was changed every day for 12 days, when rosette-like structures started to emerge. Cells were then cultured in NDM supplemented with 20 ng/mL basic fibroblast growth factor (bFGF, Peprotech) for 2–3 days to expand the rosettes. Areas with rosettes were mechanically lifted and cultured in suspension on ultra-low-attachment plates (Corning) in NDM for 2 days to allow for neural progenitor sphere (NPS) formation. Then, NPSs were maintained and expanded in NDM supplemented with 10 ng/mL bFGF and 10 ng/mL epidermal growth factor (EGF) (both from Peprotech), with media changed every other day.

#### 2.9.1. Isolation and Maturation of Neurons

For neuronal isolation, NPSs from Ctrl- and AD-patient-derived iPSCs were maintained in supplemented NDM and split manually every week for 1 month. For neuronal maturation, NPSs were dissociated with StemPro^®^ Accutase (Thermo Fisher Scientific) and plated on Poly-ornithine + Laminin-coated dishes in Neuron Induction Media (NIM) consisting of DMEM-F12 and Neurobasal (1:1), 1% N2, 2% B27 without vitamin A, 1% L-Glutamax and 0.5% penicillin/streptomycin (50 IU/50 mg/mL) (all from Invitrogen, Waltham, MA, USA), supplemented with 20 ng/mL of neuronal growth factors BDNF and GDNF (both from Peprotech), for 1 month prior to experiments. iPCS-derived midbrain dopaminergic neurons expressed Tau and microtubule-associated protein 2 (MAP-2), as well as pre- and post-synaptic markers [25]. When compared to astrocytes and microglia, SYP expression was predominant in neurons, while Cx43, TREM2 and GFAP were scarcely represented (Appendix A). Cells also exhibited synapsin-1, synaptic vesicle protein (SV-2), postsynaptic density protein (PSD-95), MAP-2 and Tau, as documented in the Results section.

#### 2.9.2. Differentiation, Maturation and Immunostimulation of Astrocytes

To obtain astrocytes from Ctrl- and AD-patient-derived iPSCs, NPSs were maintained in astrocyte differentiation medium (ADM) consisting of DMEM/F12, 1% N2, 1% Glutamax, 1% non-essential amino acids, 0.5% penicillin/streptomycin (50 IU/50 mg/mL) and 0.5 IU/mL heparin (Leo Pharma, Lisbon, Portugal) supplemented with 10 ng/mL bFGF and 10 ng/mL EGF (Peprotech), as described in [27]. The medium was changed every 2–3 days, and spheres were split manually every week for 5–7 months to ensure pure astroglial cultures. For astrocyte maturation, spheres were dissociated with StemPro^®^ Accutase (Thermo Fisher Scientific) and plated on Matrigel-coated dishes in ADM supplemented with 10 ng/mL Ciliary Neurotrophic Factor and 10 ng/mL Bone Morphogenetic Protein 4 (both from Peprotech) 7 days prior to experiments. In parallel, astrocytes cultured in ADM medium were immunostimulated with TNF-α (30 ng/mL), IL-α (3 ng/mL) and C1q (400 ng/mL) (Peprotech) for 48 h to induce astrocyte reactivity, as occurs by local sensibilization derived from activated microglia [44]. Immunostimulated and non-immunostimulated astrocytes (naïve astrocytes) were collected for RNA and protein extraction, as well as for immunocytochemistry. Their secretomes were stored for the isolation of exosomes and their further characterization. The expression of specific astrocytic proteins, such as GFAP and S100 calcium-binding protein B (S100B), as well as Cx43 gene expression, was used for astrocyte characterization, as described in [27] and documented in the Results section and Appendix A. Astrocytes evidenced low gene expression of SYP and TREM2 (Appendix A).

### 2.10. RNA Extraction and RT-qPCR

Total RNA was extracted from each of the iPSC-differentiated cells, from hippocampal organotypic cultures and from neuroblastoma cells using TRIzol^®^ reagent (Thermo Fisher Scientific) according to the manufacturer’s instructions. RNA from human CSF samples was extracted as indicated in its specific section above. Total RNA from exosomes was extracted using the miRCURY^TM^ LNA^TM^ Universal RT miRNA PCR kit (Qiagen, Venlo, The Netherlands), while the remaining exosome-free cell medium (after exosome isolation) was used for the evaluation of soluble miRNA content using the miRNeasy Serum/Plasma kit (Qiagen) according to the manufacturer’s instructions. The quantification of all samples was performed with a Nanodrop^®^ ND-100 Spectrophotometer (NanoDrop Technologies).

For miRNA determination, equal amounts of RNA were reverse-transcribed into cDNA using the Universal cDNA Synthesis Kit (Qiagen). Then, miRNA expression was determined by Real-Time Reverse Transcription-Quantitative Polymerase Chain Reaction (RT-qPCR) using the miRCURY LNA^TM^ Universal RT miRNA PCR kit (Qiagen) with predesigned miRNA LNA primers (Appendix A). For mRNA determination, the same amounts of total RNA were reverse-transcribed into cDNA using the GRS cDNA Synthesis Master Mix kit (GRiSP, Porto, Portugal), and RT-qPCR was performed using Xpert Fast Sybr Blue (GRiSP) as the master mix with specific predesigned primers (Appendix A). Both miRNA and mRNA RT-qPCR were run on the QuantStudio 7 Flex RT-PCR System (Applied Biosystems, Waltham, MA, EUA). The running conditions for miRNAs consisted of polymerase activation/denaturation and well-factor determination at 95 °C for 10 min, followed by 50 amplification cycles at 95 °C for 10 s and 60 °C for 1 min (ramp-rate 1.6 °C/s). The running conditions for mRNA determination were 50 °C for 2 min, followed by 95 °C for 2 min, 40 cycles at 95 °C for 5 s and 62 °C for 30 s. Melt-curve analysis was performed after amplification, and the specificity of PCR products was confirmed. The expression data from at least four independent experiments were processed using the 2^−ΔΔCT^ method with the internal controls glyceraldehyde 3-phosphate dehydrogenase (GAPDH) and β-actin for mRNA and U6 for miRNA, together with the exogenous control Spike-in. The miRNA results were normalized using the binary logarithm of fold change (Log_2_ FC).

### 2.11. Immunocytochemistry and Immunohistochemistry

The following primary antibodies were used: rat anti-CD11b (1:50, Biolegend, San Diego, CA, USA), rabbit anti-TMEM119 (1:50, Abcam, Cambridge, UK), rabbit anti-Iba1 (1:100, FUJIFILM Wako, Osaka, Japan), rabbit anti-CX3CR1 (1:100, Santa Cruz Biotechnology, Dallas, TX, USA), rabbit anti-MFGE8 (1:100, Santa Cruz Biotechnology), mouse anti-iNOS (1:100, BD Biosciences, Franklin Lakes, NJ, USA), goat anti-Arg 1 (1:100, Santa Cruz Biotechnology), rabbit anti-TREM2 (1:100, Cell signaling, Danvers, MA, USA), rabbit anti-synapsin-1 (1:100, produced in house), mouse anti-MAP2 (1:100, Millipore, Burlington, MA, USA), rabbit anti-Tau (1:200, Synaptic Systems, Goettingen, Germany), rabbit anti-SV-2 (1:200, Synaptic Systems), mouse anti-PSD-95 (1:200, Millipore), rabbit anti-S100B (1:200, Abcam) and mouse anti-GFAP (1:200, Novocastra, Newcastle, UK). As secondary antibodies, we used goat anti-rabbit conjugated with AlexaFluor405 (1:500), goat anti-rabbit conjugated with AlexaFluor488 (1:500), goat anti-mouse conjugated with AlexaFluor488 (1:500), goat anti-mouse conjugated with AlexaFluor594 (1:500), donkey anti-rat conjugated with AlexaFluor594 (1:500), chicken anti-goat AlexaFluor594 (1:500) and goat anti-mouse Alexafluor647 (1:500), all from Thermo Fisher Scientific. In addition, filamentous actin (f-actin) of neurons was stained using phalloidin conjugated with the AlexaFluor594 probe (1:100 in PBS with 1% BSA, Thermo Fisher Scientific).

For immunocytochemistry, cells were plated onto coverslips and fixed with paraformaldehyde (PFA) (4% *w/v* in PBS) for 20 min. Then, cells were washed with PBS and permeabilized with 0.2% Triton-X100 in PBS for 10 min. Blocking was performed with BSA at 3% in PBS for 30 min. All antibodies were diluted in PBS (1% BSA). Primary antibodies were incubated overnight at 4 °C, while secondary antibodies were incubated for 2 h at RT. Coverslips were washed by dipping them in PBS and incubated for 2 min with Hoechst 33,258 dye at 1:1000 in BSA (1% in PBS) to stain nuclei (Sigma-Aldrich). The coverslips were quickly immersed in methanol and mounted in Fluoromount-G (Thermo Fisher Scientific).

Using immunohistochemistry, we identified the transplanted neuroblastoma cells and evaluated the glial localization/reactivity in hippocampal slices through the quantification of specific cell markers, namely, Iba1 for microglia and GFAP for astrocytes. Briefly, slices were washed with PBS and immediately fixed with 4% PFA for 1 h at RT. Slices were washed with PBS and transferred from the insert mesh into a 24-well plate, 1 slice per well, followed by blocking for 3 h at RT with a solution based on Hanks’ Balanced Salt Solution (HBSS) supplemented with 10% FBS, 2% heat-inactivated horse serum, 1% BSA, 0.25% Triton X-100 and 1 nM HEPES. Hippocampal slices were incubated with primary antibodies diluted in blocking solution at 4 °C for 48 h. Then, slices were washed 3 times for 20 min with 0.01% Triton X-100 in PBS solution, followed by incubation with secondary antibodies for 24 h at 4 °C. After washing again, cell nuclei on the slice were stained with Hoechst 33,258 dye (1:1000 in BSA 1% in PBS) (Sigma-Aldrich) for 5 min. Slices were washed 3 additional times for 15 min. Finally, slices were mounted with one drop of Fluoromount-G, and one coverslip was applied to the top.

### 2.12. Confocal Microscopy and Post-Acquisition Analysis

Confocal fluorescence images of cell cultures (in mark and find mode) and of hippocampal organotypic cultures (in tilescan mode) were acquired using a Leica TCS SP8 inverted microscope (Leica Microsystems, Wetzlar, Germany) with a 40× (oil) immersion objective, sequential laser excitation at 405/488/552/638 nm and spectral detection adjusted for the emissions of AlexaFluor 405/488/594/647 dyes, respectively. The equipment was operated by Leica LAS X software (Leica Microsystems). Post-acquisition analysis, including image concatenation, z-stack analysis, fluorescence intensity quantification and localization, were performed using Fiji software [53].

### 2.13. Exosome Isolation

Exosomes were isolated from the secretomes of neurons and astrocytes by differential ultracentrifugation, as we previously described [25,38]. Equal volumes of cell media were promptly centrifuged for 10 min at 1000× *g* to pellet cell debris. Then, supernatants were transferred into new tubes and centrifuged at 16,000× *g* for 1 h to pellet and discard the large extracellular vesicles, known as microvesicles. The remaining supernatant was filtered using a 0.22 μm pore size membrane, transferred into new tubes and centrifuged for 2 h at 100,000× *g* in an Ultra L-100XP ultracentrifuge (Beckman Coulter, Brea, CA, USA) to isolate the exosomes. The pellet was resuspended/washed in phosphate-buffered saline (PBS) and recentrifuged for 2 h at 100,000× *g*. The final pellet of exosomes was resuspended in 350 μL of Qiazol (Qiagen) for RNA extraction and further evaluation of the miRNA content.

### 2.14. Evaluation of the Phagocytic Ability of Microglia

The evaluation of the microglial phagocytic capacity was assessed by measuring the fluorescence signal of internalized pHrodo™ Green Zymosan fluorescent beads (Thermo Fisher Scientific) inside the area occupied by CD11b-positive microglial cells, as previously described [54]. The method consisted of incubating beads with the microglial cells to a final concentration of 100 μg/mL for 75 min at 37 °C. Thereafter, cells were fixed with freshly prepared 4% PFA in PBS solution. Afterward, microglial cells were stained for CD11b, nuclei were counterstained with Hoechst 33,258 dye, and fluorescence images were acquired and analyzed using Fiji software.

### 2.15. Determination of Synaptic Puncta in Neurons

Synaptic protein staining was performed in iPSC-derived neurons after 77 days of differentiation and maturation. The synapsin-1 and PSD-95 puncta and their co-localized pairs were manually counted in well-defined sections of each neuronal cell using Fiji software. The synaptic puncta were determined as the number of co-localized synapsin-1/PSD-95 pairs per 10 µm of neurite length, as previously described [55].

### 2.16. Protein Extraction and Western Blot

Protein extracts from cell lysates were obtained using the TripleXtractor reagent (GRiSP, Portugal) following the manufacturer’s recommendations. To measure the release of soluble APP beta (sAPPβ) and of high mobility group box 1 (HMGB1) protein, total proteins were precipitated from the cell culture medium using trichloroacetic acid in 10% (*v*/*v*) acetone (9:1), washed in cold acetone containing 20 mM Dithiothreitol and solubilized in sample buffer (1:1 Urea 8M (in Tris HCl 1M, pH 8) and 1% (*v*/*v*) SDS). Then, protein extracts were sonicated for 15 s (in a maximum of 5 cycles, 80% amplitude, 90% pulse) and centrifuged at 3200× *g* for 10 min at 4 °C. Protein concentration was determined using the NanoDrop^®^ ND-100 Spectrophotometer (NanoDrop Technologies). Equal amounts of protein were separated by Tris-Tricine gel electrophoresis for ~1:30 h at 200 V and 100 mA. After electrophoresis, proteins were transferred onto nitrocellulose membranes for 1:30 h at 200 V and 300 mA. Membranes were blocked with 1% Tween 20-Tris-buffered saline (T-TBS) plus 5% (*w*/*v*) non-fat dried milk for 1 h at RT with shaking and incubated overnight at 4 °C with the primary antibodies rabbit anti-GFAP (1:500, Sigma-Aldrich), rabbit anti-S100B (1:750, Dako-Agilent, Santa Clara, CA, USA), rabbit anti-APPβ (1:500, IBL, Fujioka, Japan), mouse anti-HMGB1 (1:200, BioLegend) or mouse anti-β-actin (1:5000, Sigma-Aldrich) diluted in T-TBS with 5% BSA. After washing, membranes were incubated for 1 h at RT (with shaking) with goat anti-mouse HRP (1:5000) or goat anti-rabbit HRP (1:5000), both from Santa Cruz Biotechnology, diluted in blocking solution. Immunoreactive bands were detected using the WesternBright Sirius HRP substrate (Advansta, San Jose, CA, USA) and visualized using ChemiDoc™ equipment (Bio-Rad, Hercules, CA, USA). Densitometric analysis of protein bands was performed with Image Lab™ analysis software (Bio-Rad). The results from cell lysates (GFAP and S100B) were normalized to β-actin, and the results from extracellular media (sAPPβ and HMGB1) were normalized to Amido Black total protein stain, as per usual procedures in our lab [56].

### 2.17. Soluble Cytokine Determination

To evaluate the release of soluble cytokines, such as TNF-α and monocyte chemoattractant protein-1 (MCP1), as well as interleukin (IL)-6 and IL-18 into cell media, we used the LEGENDplex multiplex immunoassay (BioLegend) according to the manufacturer’s instructions. Data were recorded on a Guava easyCyte 5 HT flow cytometer (Millipore), operated by Guava Nexin Software, and further processed using LEGENDplex™ Data Analysis Software V8.0 (BioLegend, San Diego, CA, USA). To determine the S100 calcium-binding protein B (S100B) concentration in the extracellular medium, we used an enzyme-linked immunosorbent assay (ELISA), as we described in [57].

### 2.18. Statistical Analysis

Statistical differences in miRNA expression between the MCI-Ctrl and MCI-AD groups were determined by the Mann–Whitney U non-parametric test, considering that the CSF miRNA dataset does not follow a normal distribution, as assessed by Shapiro–Wilk (α = 0.015) and Kolmogorov–Smirnov (α = 0.031) tests. The requirements for the use of the Mann–Whitney U test were met, i.e., the existence of random and independent samples and a continuous data scale. Bonferroni post hoc correction was used to test multiple comparisons, and only values of *p* < 0.05 were considered significant.

The remaining pairwise comparisons between each control group and the respective AD group (non-/SH-transplanted hippocampal slices vs. SWE-transplanted hippocampal slices; MG-Ctrl vs. MG-AD; Neu-Ctrl vs. Neu-AD; and Ast-Ctrl vs. Ast-AD—including naïve vs. immunostimulated) were all performed using two-tailed Student’s *t*-test assuming unequal variances, and only differences with *p* < 0.05 were considered significant. Statistical analysis was performed using GraphPad PRISM 9.0.0 software (GraphPad Software Inc., San Diego, CA, USA).

## 3. Results

### 3.1. SH and SWE Human Neuroblastoma Cells Were Successfully Transplanted into 4 DIV Organotypic Hippocampal Cultures from WT Mice

Organotypic hippocampal cultures (HCs) have been used as a model to assess neurodegeneration, neurotoxicity and neuroprotection, including inflammatory responses [46,58]. Recently, the maturation, integration and therapeutic benefits of human induced neural precursor/stem cells were assessed in hippocampal slices [59]. Our previous studies showed that miR-124, miR-125b, miR-21, miR-146a and miR-155 were upregulated in SWE cells, together with higher APP and Aβ production, when compared to SH cells, and that miR-124 drove microglia polarization [25,39,40]. Here, our goal was to identify whether the transplantation of SWE cells in HCs (Figure 1A), used as an AD cell model, was able per se to induce microglial and astrocytic activation, as well as changes in homeostatic miRNA imbalance in an integrated system.

To identify the transplanted cells in HCs and to monitor their engraftment, SH and SWE neuroblastoma cells were stained with the cell tracker, a live-cell dye suitable for monitoring cell movement or location. Before transplantation, we monitored how long the cell tracker remained stable in the transplanted cells (Figure 1B). The results showed that while the cell tracker was stable for 4 DIV, it significantly lost fluorescence intensity (*p* < 0.01, more than 50%) if used for 7 DIV (Figure 1C). According to these data, we decided to maintain the transplanted HCs for 4 DIV, enabling the tracking of the engrafted SH/SWE cells. Therefore, each mouse HC (400 μm thick) was incubated with 1 × 10^4^ SH or SWE neuroblastoma cells, according to a previous study [46]. As observed in Figure 1D (full-size culture) and in Figure 1E (close-up sections), both SH and SWE cells were successfully transplanted into HCs after 4 DIV. No significant differences were found in the transplantation rate of SH vs. SWE cells, with both cell types showing more than 80% cell-tracker-positive cells engrafted in the hippocampal margins (indicated by bold dashed lines), though not uniformly distributed.

### 3.2. Hippocampal Slices Transplanted with SWE Cells Show Microglial Activation and Astrocyte Reactivity, Together with miR-21 Overexpression

After maintaining the HCs for 4 days post-transplantation with SH (HC-SH) or with SWE cells (HC-SWE), nuclei staining showed semi-organized hippocampal coronal sections (Figure 2A, dashed line contour). Most of the hippocampal tissue was covered by cell-tracker-positive SH or SWE cells, demonstrating successful transplantation. An increased number of Iba1+ and GFAP+ cells was observed after the transplantation of SWE cells in HCs in comparison with the non-transplanted (NT) or HC-SH slices, where only some reactive microglia were noted (Figure 2A).

Representative close-up sections selected from similar hippocampal regions (nearby *Cornu Ammonis 3* (*CA3*)) depict some of the major density differences found in hippocampal microglia and astrocytes (Figure 2B,E), mainly after SWE cell transplantation. Stimulation was markedly higher in HC-SWE slices relative to HC-SH slices for Iba1+ cells (*p* < 0.01) or Iba1 FI (*p* < 0.05) or as compared with NT slices (*p* < 0.01) in both conditions (Figure 2C,D). Data evidence the surveillance role of microglia and the alert condition upon pathological SWE cell interaction. Regarding astrocyte reactivity induction, despite the small increase in the number of GFAP+ cells (Figure 2F, *p* = 0.07), marked differences were obtained in relative GFAP FI as compared with either NT or HC-SH assays (Figure 2G, *p* < 0.01 for both cases), emphasizing the stress reactivity of astrocytes to SWE cell xenotransplantation. Data support that organotypic hippocampal slices are a good model to gain insight into the mechanisms associated with SWE-cell-induced gliosis and neuroinflammation. Thus, we then investigated possible changes in the miRNA signature after hippocampal transplantation with each type of cell (SH or SWE cells). Transplantation with SWE cells, but not with SH cells, clearly upregulated miR-21 (Figure 2H, *p* < 0.01 vs. NT or HC-SH slices). No other significant alteration in the assessed miRNAs were produced by the transplanted SH or SWE cells. Data highlight that SWE cells, in addition to polarizing astrocytes and microglia, lead to upregulated miR-21 in HCs. Whether miR-21 is dysregulated in the CSF of AD patients and the contribution of each cell type to its upregulation was next investigated.

### 3.3. miR-21 Upregulation in the CSF Discriminates MCI-AD from MCI-Ctrl Patients

Our previously published data have shown the upregulation of miR-21 in SWE cells and in iPSC-derived neurons from patients with the *PSEN1* mutation [25,39,40]. Though the expression of miR-21 is not consistent in the different AD models, most of them support its upregulation [34], which is consistent with the notorious elevation of miR-21 in hippocampal slices transplanted with SWE cells. However, miR-21 has been scarcely explored as a potential biomarker in the CSF of AD individuals. To gain insight into this issue, we benefited from a small collection of CSF samples from a series of MCI-AD patients, where we assessed miR-21 representation and that of others equally associated with neuroinflammation.

CSF samples were collected from patients who fulfilled the criteria adopted for MCI due to AD (MCI-AD), according to the National Institute on Aging—Alzheimer’s Association workgroups established by Albert et al. [5]. Briefly, MCI-AD patients showed positive PiB-PET, low CSF Aβ42 and high CSF levels of total and phospho-Tau (Table 1). As controls, we used CSF samples from a similar number of subjects with clinical and cognitive criteria for MCI, but without AD-related biomarker alterations (MCI-Ctrl) (Figure 3A). The data revealed that only miR-21 was significantly increased in the MCI-AD group when compared with the MCI-Ctrl one (Figure 3B, *p* < 0.01). When further miRNAs considered to be also important in AD were explored no other changes were found (Appendix A). It should also be noted that miR-21 was the one that showed less inter-individual variability relative to the others, including miR-124, miR-125b, miR-146a and miR-155 (Figure 3), as well as miR-23a, miR-29a, miR-122, miR-219a and miR-338 (Appendix A), which all revealed high inter-individual variability, particularly in the MCI-Ctrl group. Despite the low number of patients under analysis, our data suggest a lower inter-individual variability in miR-125b, miR-155, miR-122 and miR-138 in the CSF from MCI-AD patients.

Therefore, our data indicate that miR-21 is involved in the pathogenesis of AD, but it is necessary to understand whether its overexpression is associated with a particular neural cell type before it may be considered an accurate therapeutic target. In this context, we decided to next explore the expression of miR-21, together with that of miR-124, miR-125b, miR-146a and miR-155, in microglia, neurons and astrocytes differentiated from AD patient iPSCs generated from their fibroblasts.

### 3.4. MG-AD Cells Show Dysregulated Immunoreactive Markers and Upregulated miR-146a and miR-21

Microglial miR-21 expression was shown to increase after hypoxic treatment and to prevent neuronal death [34], but also to decrease after the expression of mutant human SOD1 in N9 murine microglia [56]. On the other hand, the elevation of miR-21 led to microglial pro-inflammatory polarization in diverse pathological contexts [39,60,61,62]. Our previous study using human SH/SWE neuroblastoma cells and CHME3 microglia cocultures revealed that increased miR-21 in SWE cells was secreted into their exosomes, which led to CHME3 microglial miR-21 overexpression and its inclusion in their derived exosomes [39]. The data in Figure 2 show an increased number of Iba1+ cells and enhanced Iba1 fluorescence intensity, together with the elevation of miR-21, in hippocampal slices transplanted with SWE cells. Thus, having the opportunity to work with microglia derived from *PSEN1-*mutant iPSCs generated from AD patients (MG-AD), here, we explored whether the pathological microglia overexpressed miR-21 or other inflammatory-associated miRNAs when compared to matched Ctrl cells (MG-Ctrl) (Table 2).

AD and Ctrl MG cells were generated using a recently described protocol based on primitive hematopoiesis [26], as schematized in Figure 4A, comprising a series of sequential steps, illustrated in Appendix A.

Upon differentiation, we obtained microglia revealing markers and phagocytic ability, as reported by Konttinen and colleagues [26] (Figure 4B,C). We confirmed the ubiquitous expression of CD11b, TMEM119, Iba1 and CX3CR1, together with the MFGE8 protein, known to participate in microglial phagocytosis [63,64]. This ability was confirmed through the incubation of microglia with pHrodo™ Green Zymosan fluorescent beads (Figure 4C). Panel D depicts the percentage of cells that engulfed specific numbers of beads. Nearly 45% of the microglial cells showed internalized beads. Of those, 23.5% presented 1–2 internalized beads, 13.3% presented at least 2–3 beads, and the remaining 7.5% presented more than 3 beads.

Next, we compared the expression of immune regulators and cytokines in MG-AD with those exhibited by MG-Ctrl cells. Although no significant changes were detected for TGF-β, IL-10, IL-6, MHC-II, CX3CR1 and CD68 (Figure 4E), an increased immunofluorescence signal was found for iNOS (*p* < 0.001), with a decrease in Arg-1 (*p* < 0.01) and TREM2 (*p* < 0.05) in MG-AD vs. MG-Ctrl (Figure 4F,G). Data from RT-qPCR confirmed the downregulation of Arg-1 vs. MG-Ctrl cells (*p* < 0.01, Figure 4H), while iNOS did not reach significance, and TREM2 showed upregulated levels. When our selected miRNAs were evaluated, we identified the upregulation of miR-21 and miR-146a in MG-AD vs. MG-Ctrl cells (Figure 4I, *p* < 0.05), supporting microglial immune-regulation alterations in AD.

### 3.5. Neu-AD Cells Exhibit Upregulated Intracellular and Exosomal miR-21 and miR-124, Together with miR-125b in Cells and miR-155 Only in Exosomes, and Sex-Biased miR-21 Expression

Earlier studies similarly demonstrated that miR-21 plays an important role in AD neurons by moderating their apoptosis upon Aβ exposure [35]. Our previous study further confirmed that miR-21 is upregulated in both differentiated SWE neuroblastoma cells and AD patient iPSC-differentiated neurons [25]. Using our optimized protocol [25], we successively differentiated and matured iPSC-derived midbrain neurons (Neu), as schematized in Figure 5A. We used the AD and Ctrl cell lines indicated in Table 2, including the same samples used for studies with microglia.

After 77 days in culture for neuronal differentiation and maturation, we observed the typical neuronal cytoskeletal organization using MAP-2 and Tau family proteins, as well as the synaptic markers synapsin-1, SV-2 and PSD-95 (Figure 5B). In addition, we also observed f-actin staining in neurons, a key component of both dendrites and spines. Further analysis of the pre- and post-synaptic puncta with synapsin-1 and PSD-95, respectively, and their co-localization are presented in Figure 5C,D. Although the results show that synapsin-1 puncta were more abundant than PSD-95 puncta (3.4 ± 0.7 vs. 2.2 ± 0.5, respectively), nearly 84% of the PSD-95 puncta co-localized with synapsin-1 puncta.

The assessment of MAP-2, Tau, PSD-95 and SV-2 expression levels did not reveal differences between AD and Ctrl samples (Figure 5E). However, there was a clear upregulation of miR-21, miR-124 and miR-125b (*p* < 0.05 for all) in Neu-AD cells when compared to Neu-Ctrl cells. We have previously shown that exosomal miRNA cargo may depend on intracellular miRNA levels and active/passive transference [39,61,62]. Consistently, exosomes recapitulated the Neu-AD levels of miR-21 and miR-124 (*p* < 0.05 for both), but not those of miR-125b (Figure 5G). Notably, exosomal miR-155 upregulation contrasted with the relatively low levels of miR-155 in cells. Such increased levels of miR-155 may be associated with the predisposition to neuroinflammation and its dissemination. According to recent studies, there are sex-related susceptibilities to developing AD and immune responses [65]. Intriguingly, miR-21 levels were revealed to be higher in the male sample compared to the female specimen, both in cells and their derived exosomes (*p* < 0.01 for both), while no differences were noted for the other miRNAs in our analysis. Though further studies are required to validate the data on exosomes from AD neurons, our data suggest that miR-21 (mainly in male patients), miR-124 and miR-155 have the potential to be considered promising biomarkers of neuronal AD pathology in circulating exosomes.

### 3.6. Ast-AD Cells Show Phenotypic Diversity, GFAP+ Cell Deficiency, Altered Inflammatory Gene/miRNA Expression and Contrasting Cellular/Exosomal miR-21 Profile

Like microglia and neurons, astrocytes are included in the catalog of cells whose properties are regulated by inflammatory-associated miRNAs, including miR-21. Among other functions, this miRNA was demonstrated to be involved in astrocytic hypertrophy, glial scar progression and astrocyte reactivity [66,67].

Astrocytes (Ast) from the AD and Ctrl cell lines that were used to obtain MG and neurons (Table 2) were generated after 157 days following the astroglial differentiation protocol [27,51] (Figure 6A). The cells displayed heterogeneous shapes, but mostly the stellate morphology characteristic of glial cells [27] (Figure 6B), and the astrocyte-associated GFAP and S100B markers (Figure 6C), with no differences in immune fluorescence intensity between Ast-AD and Ast-Ctrl (Figure 6C). However, when evaluated for the number of GFAP-positive cells, Ast-AD exhibited a lower number compared to Ast-Ctrl (75.6% vs. 91.2%, *p* < 0.05), but their numbers were similar if considering S100B-positive cells (98.2% vs. 99.9%, Figure 6D). As previously mentioned, cell proliferation differences were not found, attesting the lower number of GFAP+ cells in the pathological astrocytes.

Morphologically, considering the cell perimeter (yellow outline) and the cell soma (blue outline) (Figure 6E), we identified four major classes, as proposed by Jones et al. [28]: arborized, polarized, fibroblast-like and rounded shapes. The arborized morphology was present in more than 80% of all astrocytes. No significant changes were observed between Ast-Ctrl and Ast-AD cells in any of the morphological subtypes (Figure 6F).

When assessed for differences in inflammatory gene expression between Ast-AD and Ast-Ctrl (Figure 6G), we obtained lower expression levels for HMGB1 and IL-10 in Ast-AD as compared to Ast-Ctrl (*p* < 0.001 and *p* < 0.05, respectively, Figure 6G). Additionally, we found higher expression of the CCAAT/enhancer-binding protein α (C/EBPα) (*p* < 0.05) and IL-6 (*p* < 0.01) in Ast-AD vs. Ast-Ctrl cells. When the cell secretome was assessed for changes in cytokine release (TNF-α, IL-6, MCP1 or IL-18) between AD and Ctrl astrocytes, no changes were identified.

Then, as performed for microglia and neurons, we analyzed the miRNA profiles of Ast-AD cells vs. Ast-Ctrl cells for their intracellular and exosomal contents (Figure 6I,J). We found the overexpression of miR-21 (*p* < 0.01) together with the downregulation of miR-155 (*p* < 0.05) in Ast-AD cells relative to Ast-Ctrl. Intriguingly, except for miR-125b, all the assessed miRNAs were diminished in Ast-AD exosomes (at least *p* < 0.05). This is a new finding if we compare it to MG-AD and Neu-AD, indicating that miR-21 is preferentially retained in Ast-AD, causing its depletion in their exosomes. No sex-related differences were found in either AST-Ctrl or AST-AD (data not shown).

### 3.7. Immunostimulation Drives S100B Imbalance and Morphological Changes in Ast-AD

Astrocyte hypometabolism has been described to be associated with neuronal dysfunction, age, and disease [68,69]. Thus, in some disease models, such as AD, because astrocytes are known to contribute to neuroinflammation when reactive, some authors have stimulated astrocytes with TNF-α and IL-1β to make their pathological profiles more visible [27]. In the present study, we used IL-1α + TNF-α + C1q to stimulate Ast-Ctrl and Ast-AD cell neurotoxicity [44] (Figure 7A). This sensitization intends to recapitulate that caused by microglial activation, a current condition in AD [70,71], reported to enhance GFAP protein levels and astrocyte neurotoxicity [44].

As such, we first focused on GFAP and S100B changes upon immunostimulation. As shown in Figure 7B, no major alterations were observed in Ast-AD relative to Ast-Ctrl in either the mean fluorescence intensity (Figure 7C) or the number of GFAP- and S100B-positive cells (Figure 7D). Curiously, the low number of GFAP-positive cells previously observed in naïve Ast-AD cells (Figure 6D) was maintained after exposure to stress conditions (*p* < 0.01, Figure 7D). To further validate these data, we additionally explored GFAP protein expression by Western blot of the pooled lysate samples from Ast-Ctrl and Ast-AD samples in the absence and presence of the immunostimulation process (Figure 7E,F). The treatment did not modify the decreased intensity of GFAP bands in either naïve or stressed Ast-AD cells. The assessment of S100B levels by Western blot (Figure 7E,G) revealed that non-stimulated Ast-AD cells had lower levels than their Ast-Ctrl counterparts (*p* < 0.001). However, immunostimulation led to a decrease in S100B expression in Ast-Ctrl (*p* < 0.05) but led to its increase in Ast-AD cells (*p* < 0.05), thus better differentiating the astrocyte phenotypes associated with the AD condition.

Astrocyte morphometric alterations were also highlighted by immunostimulation. Stressed Ast-AD cells showed a dramatic reduction in the arborized morphology vs. non-stimulated cells (Figure 7H; *p* < 0.001) or matched Ast-Ctrl cells (*p* < 0.01). These alterations resulted in a significant increase in fibroblast-like (*p* < 0.01) and rounded morphological phenotypes (*p* < 0.05), causing a major reduction in the mean cell perimeter of stressed Ast-AD (Figure 7I; *p* < 0.001) vs. respective naïve and vs. stressed Ast-Ctrl cells (*p* < 0.05). Consequently, a significant increase in the mean cell surface area was observed in the stressed Ast-AD cells (Figure 7J) relative to non-treated (*p* < 0.05) or stressed Ast-Ctrl cells (*p* < 0.01).

### 3.8. Immunostimulated Ast-AD Cells Show Increased Inflammatory Gene Expression, Exosomal Enrichment in miR-21 and Release of sAPPβ and Cytokines

To further assess the contribution of Ast-AD when immunostimulated to the abnormal amyloid status and inflammatory signaling molecules, we compared the release of sAPPβ (a product of the amyloidogenic pathway) by Ast-AD and Ast-Ctrl cells in the absence and in the presence of C1q + IL-1α + TNF-α, as schematized in Figure 8A. The release of sAPPβ was markedly and exclusively enhanced upon immunostimulation in Ast-AD cells vs. the other conditions (Figure 8B,C; *p* < 0.05), thus potentially contributing to its dissemination and harmful consequences in AD. We found that IL-10, TNF-α and IL-8 gene expression was also enhanced in Ast-AD cells (Figure 8D; *p* < 0.05) as compared to matched Ast-Ctrl cells. This finding indicates the over-reactive response of Ast-AD cells relative to that of Ast-Ctrl cells, with noticeable consequences for the secretome composition.

As far as the release of HMGB1 is concerned, a critical alarmin whose dysregulation has been implicated in astrocyte dysfunction in the neurodegenerative context [72], naïve Ast-AD cells showed a reduced release when compared with their Ast-Ctrl counterparts (Figure 8E,F; *p* < 0.001), validating our previous data on HMGB1 gene expression (see Figure 6G). However, HMGB1 levels from both Ast-Ctrl and Ast-AD were significantly boosted upon immunostimulation (*p* < 0.001 and *p* < 0.05, respectively), though the stressed Ast-AD cells were confirmed to have a lower response to the tested stimuli, suggesting higher immune tolerance. Nevertheless, we observed the prompt release of TNF-α, IL-6, MCP1 and IL-18 by immunostimulated Ast-AD (Figure 8G; at least *p* < 0.05), which was, in some cases, higher (at least *p* < 0.05 for the first three markers) than that in the corresponding Ast-Ctrl samples. When compared to the average levels released by the same cells under naïve conditions (dashed lines), this release was, in some cases, 2-to-3 orders of magnitude elevated. These results suggest that inflammatory conditions may precipitate and perpetuate the astrocyte-induced homeostatic imbalance in AD pathology.

Therefore, we wondered whether the immunostimulation of Ast-AD caused significant changes in the exosome cargo dynamics, especially in exosomal miR-21. While no alterations were observed in cells treated with the C1q + IL-1α + TNF-α cocktail (Figure 8H), there was clear enrichment in the exosomal miR-21 cargo with the immunostimulation of Ast-AD (Figure 8I; *p* < 0.01), contrasting with the previous data on untreated cells (Figure 6J). Therefore, the asymmetry in exosomal miR-21 between naïve and immunostimulated Ast-AD cells may be relevant for further consideration in AD, especially aiming at the dynamic modulation of brain immune responses and disease propagation through paracrine communication.

### 3.9. miR-21 Upregulation Suppresses its PPARα Target in SWE-Transplanted Hippocampal Slices and in AD iPSC-Derived Cells Depending on the Concurrent Dysregulated miRNAs

miR-21 was shown to target peroxisome proliferator-activated receptor alpha (PPARα) [73] and phosphatase and tensin homolog (PTEN) [74,75], among others. To explore how much the upregulation of miR-21 in our different models was able to modulate these direct targets, we evaluated their expression levels in the different AD models and the respective controls used in the present study, from the organotypic hippocampal cultures transplanted with SWE cells (or SH) (Figure 9A) to the microglia, neurons and astrocytes differentiated from AD patient iPSCs or matched controls (Figure 9C).

The data on the transplanted organotypic hippocampal clearly indicate that PPARα, but not PTEN, was downregulated after transplantation with SWE cells for both murine and human genes (*p* < 0.01 vs. NT or HC-SH slices, Figure 9B), reinforcing the relevance of the upregulated levels of miR-21 in governing the aberrant glial phenotypes in this AD model. It is emphasized that miR-21 was the only upregulated one among our assessed miRNAs. When the same assessments were performed in MG differentiated from patient iPSCs, an identical result was observed, with the strong downregulation of PPARα only in MG-AD overexpressing miR-21 (Figure 9D). Note that these cells also expressed elevated miR-146a, which seems to not counteract the effects of miR-21 on its target. Similar results were not verified in Neu-AD, where PPARα was not found to be downregulated, despite the increased expression of miR-21. We may hypothesize that the concomitant elevation of miR-124 and miR-125b may have abolished this suppression. The evaluation of PPARα in Ast-AD was not what we anticipated. Despite the elevation of miR-21 in cells, we were not able to observe a decrease in PPARα gene expression. We hypothesize that the simultaneous downregulation of miR-155 observed in Ast-AD may have contributed to PPARα elevation in these cells (*p* < 0.05). It may indeed be the case that, once immunostimulated, defective miR-155 was no longer present, which, together with the formation of exosomes enriched in miR-21, might have driven PPARα downregulation. Further studies are required to dissect the intricate effects of such combinatorial inflammatory miRNAs on their cellular targets.

## 4. Discussion

At present, there is no effective therapy for AD, which is why it is essential to explore new potential targets for the development of promising target-driven therapies. Accumulating evidence shows that miRNAs may play a critical role in the pathogenesis of AD [76,77,78]. Nevertheless, changes in miRNA profiles in human models of AD have been scarcely investigated, and translation to the regulation of miRNAs as a potential therapy is lacking. Therefore, it is highly relevant to accelerate research on brain cells and exosomal miRNAs, mainly those that have been associated with inflammation, a condition associated with AD predisposition, spread and pathology [79,80]. Among the several dysregulated miRNAs, miR-124, miR-125b, miR-146a, miR-155 and miR-21 have been the most highlighted miRNAs in biological fluids and biopsy specimens of people with AD [80,81].

Our previous studies identified elevated levels of miR-124, miR-125b and miR-21 in *PSEN1* mutant iPSC-derived neurons and SH-SY5Y-*APP695 Swedish* neuroblastoma cells [25,39]. Such alterations in these inflammatory-associated miRNAs and their modulations are important for recovering dysregulated biological functions in target cells toward regenerative medicine, as we demonstrated [41,82,83,84].

In this study, we identified miR-21 as having translatable potential for its consistent overexpression in different patient samples, multiple disease models and multiple cell types and its trafficking into exosomes. Here, we assessed, for the first time, how the neurons and glial cells of mouse hippocampal slices responded to the transplantation of SWE cells in terms of gliosis and changes in the above-indicated set of miRNAs. The transplantation of human iPSCs in mouse hippocampal organotypic cultures previously demonstrated that these neurons became differentiated and are anatomically integrated [46]. In another study, grafted human induced neural precursor cells rescued cognitive deficits in 5xFAD mice by reinforcing local neural circuitry [59]. Organotypic hippocampal and coculture systems have also been used to examine the pathophysiology of brain diseases and regenerative mechanisms upon pharmacological modulation, given the unique advantages of replicating many aspects of the in vivo brain [85]. For example, chronic ethanol treatment was shown to promote abnormal synaptic transmission in organotypic hippocampal slice cultures through glutamate-receptor-mediated neurotoxicity [86]. Importantly, organotypic HCs have also been successfully used to investigate the pathophysiology of AD [87,88] and as a platform to study neuroinflammation and characterize dynamic phenotypic changes in microglia upon exogenously applied stimuli [89]. Here, we observed the efficient transplantation of SWE cells, which caused increased hippocampal microglial activation when compared to non-transplanted or SH engraftment, further confirming our previous studies centered on the paracrine influence of SWE cells on microglial activation [39,40]. Similarly, mouse hippocampal astrocytes also showed increased GFAP immunofluorescence intensity, commonly considered a direct measure of astrocyte reactivity [90].

SWE cells, the most utilized model in neurodegenerative research, have contributed to advancing our understanding of AD-associated pathology [91]. We used retinoic acid to improve differentiation and favor neuron-like morphology and function, together with the expression of neuron-specific markers, conditions that are useful to evaluate cellular neurotoxicity [92]. The advantages of the model include the capacity for large-scale expansion prior to differentiation, which synchronizes the cell cycle to produce a homogeneous population, being easy to culture, low-cost and free of ethical concerns [25,93,94]. However, we should not forget that this cell line is a neuroblastoma derivative and therefore has cancerous properties that influence metabolic properties and growth performance [95]. Even so, miR-21 elevation exclusively in SWE cell-transplanted hippocampal cultures may suggest a specific response to SWE cell-induced toxicity, as previously described [35]. miR-21 was also found to be deregulated after Aβ treatment in murine hippocampal neurons [32]. Therefore, although premature, our results seem to support that miR-21 may be an early indicator of amyloid toxicity in the hippocampus.

Among the several assessed inflammatory-associated miRNAs, miR-21 was unique in revealing a marked increase in its levels in hippocampal slices transplanted with SWE cells. This finding prompted us to investigate whether miR-21 could be of diagnostic value in the CSF of AD patients, as proposed for patients with multiple sclerosis [96]. Circulating miR-21 levels showed a strong correlation with the neurofibrillary tangle score in AD patients [36], and its content in plasma-derived extracellular vesicles was proposed to discriminate between AD and dementia with Lewy bodies (DLB) [97]. However, despite its high abundance in the CSF of dementia patients (including AD), miR-21 did not discriminate AD from non-AD patients [98]. Here, despite the low number of samples, we report a clear CSF miR-21 elevation in MCI-AD patients that was not found in MCI-Ctrl individuals. In fact, our study has critical differences from the previous report. First, the study conducted by Sørensen et al. used a different criterion (NINCDS-ADRDA) for the clinical diagnosis of AD and enrolled patients with different Aβ42 and Tau profiles [99]. In turn, our study included patients in the initial AD stages, which is relevant for controlling for nonspecific lifestyle changes and advanced-stage-associated comorbidities. In addition, the detailed and rigorous characterization of AD patients according to Albert et al.’s 2011 criterion is a key point of our study [5]. It is also emphasized that all MCI patients enrolled in this study revealed similar cognitive complaints, only showing distinctive profiles in amyloid and neurodegenerative-associated biomarkers.

The consistent elevation of miR-21 in the CSF and in hippocampal slices after SWE cell transplantation is known to exhibit increased miR-21 levels [25] concomitantly with an increased number of Iba1+ and GFAP+ cells, which led us to question which cells, neuronal or glial, contributed more or were in its origin. For this purpose, we decided to use neurons, astrocytes and microglia differentiated from AD patient iPSCs as models that closely mimic their intrinsic cellular alterations in AD [99]. In fact, the development of human iPSC-based technologies provided an unprecedented opportunity to model cellular disease phenotypes in patient-derived cells, tackling the incompatibilities between animal models and humans [99,100,101]. In AD, despite their limited clinical applications, iPSC-derived brain cells hold considerable promise for understanding this disease, aiming at the identification of novel targets and better innovative treatments [93]. In this study, we differentiated iPSCs from symptomatic and pre-symptomatic AD patients carrying the *Finnish PSEN1ΔE9* familial mutation and from healthy controls and successfully obtained microglia, neurons and astrocytes, using specific protocols for each one [25,26,27]. Specific markers and gene expression levels supported the specificity of each differentiation. In particular, in the case of neural differentiation via dual SMAD inhibition [51,52] used to generate both neurons and astrocytes, minimal cross-contamination was identified [102]. Since neurons are generated first, they were isolated from NPCs after 77 days, while astrocytes were isolated only after 5–6 months (157–180 days). Such a temporal gap is reported to ensure minimal neuronal contamination in astrocyte cultures, as previously demonstrated for the same cells by Oksanen and colleagues [27]. Less than 15% SYP expression was detected in astrocyte cultures, in contrast to neuronal cultures, where almost undetectable GFAP expression was obtained.

Since the recent development of differentiation protocols to obtain microglia (or microglia-like cells) from iPSCs, several studies have emerged that used them to examine the effects of AD-linked mutations or to identify new risk factors [21,26,103,104]. Here, we generated microglial cells with the typical microglial signature, including microglia-specific markers (CD11b, TMEM119, Iba1 and CX3CR1), which also demonstrated phagocytic properties, in accordance with similar studies [21,26]. The increased iNOS and decreased Arg-1 found in MG-AD are consistent with a pro-inflammatory status, as previously documented in AD microglia [105,106]. While iNOS is known to be stimulated in AD microglia, generating nitric oxide (NO) and causing neuronal apoptosis [106], its substrate competitor Arg-1 promotes the microglial clearance of Aβ during neuroinflammation [107]. Conversely, TREM2 is an AD-associated risk gene coding a transmembrane protein that is highly expressed in microglia and known to regulate their function in neurodegeneration [104,108]. Here, its decreased immunofluorescence and increased gene expression in MG-AD suggest TREM2 imbalance, as Konttinen and colleagues reported for the same cells [26]. Together with the inflammatory status of MG-AD, we found elevated miR-21 and miR-146a levels. miR-146a upregulation was recently associated with defense mechanisms in AD [109] and miR-21 indicated to alleviate neuroinflammation, cognitive deficits and pathological changes in APP/PS1 mice [34]. Note that miR-21 overexpression mirrored what we have previously reported in the CHME3 microglial cell line exposed to SWE-derived exosomes [39]. This was expected, since miR-21 has been described as a critical immune regulator in diverse pathological contexts [60,110]. It is noted, however, that even considering that microglial cells were active contributors to miR-21 upregulation in our experimental models, it was not enough to counteract neuroinflammation and several AD-related pathological markers.

The other cell type of interest in our study was iPSC-derived neurons, reported as highly functional but also as able to recapitulate AD’s most common features, including abnormal Aβ processing and production, Tau hyperphosphorylation, altered spine density and miRNA dysregulation [25,111,112,113]. Here, we identified typical cytoskeletal and synaptic proteins (and their co-localization) in neurons, but we did not find major changes between Neu-Ctrl and Neu-AD cells for hallmarks that may derive, at least in part, from their “age resetting” reprogramming process [114]. However, Neu-AD showed upregulated miR-21, miR-124 and miR-125b, which are all found to be dysregulated in AD models [25,39,76]. Of those, only miR-21 and miR-124 were recapitulated in their exosomes, suggesting their ability to be disseminated to other cells. This is important because while the neuronal overexpression of miR-21 was shown to be neuroprotective [115,116], neuron-derived exosomes with high miR-21 content induced neurotoxicity and microglial pro-inflammatory activation [61,62]. Despite the underpowered data pointing to a sex-biased miR-21 expression pattern in Neu-AD cells, this is not without precedent since the male-dominant expression of miR-21 is already associated with an increased susceptibility to other diseases [117].

Other cells that contribute to neuroinflammation and may be implicated in the dysregulation are astrocytes, mainly when acquiring a reactive phenotype and causing excitotoxicity, the loss of synaptic plasticity and calcium dysregulation [118]. iPSC-derived astrocytes from AD patients showed the enhanced release of Aβ42, altered calcium homeostasis, increased ROS production and inflammatory cytokine release, among other features [27,119]. These cells also reveal morphological changes (heterogeneity and atrophy), as well as the aberrant expression and sub-cellular localization of S100B [28]. In our study, the continuum of GFAF/S100B-positive cells, ranging from roundish/polarized ones to highly arborized cells, agrees with previous studies [27,28]. Ast-AD revealed fewer GFAP-positive cells than S100B-positive ones, with low HMGB1 and IL-10 and elevated IL-6 and C/EBPα gene levels, already pointing to some altered properties [27,72,120]. The cells also evidenced upregulated miR-21 and downregulated miR-155, and their exosomes were defective in miR-21, miR-124, miR-146a and miR-155. The depletion of miRNAs in exosomes was also found in primary astrocytes from the *SOD1G93A* mouse ALS model [121] and may indicate defective transference from cells to their exosomes, triggering the compromise of targets in recipient cells. When dealing with astrocytes in AD, many authors have used cell immunostimulation to reveal their pathological phenotype [27,44]. Here, we used the C1q + IL-1α + TNF-α cocktail, first described by Liddelow and colleagues as inducing a reactive and neurotoxic phenotype caused by pro-inflammatory microglia, previously identified in the brains of AD patients [44]. Such immunostimulated Ast-AD cells, in addition to exhibiting marked morphological aberrancies, revealed elevated S100B protein levels and IL-10, TNF-α and IL-8 transcripts. Stressed Ast-AD similarly released increased amounts of sAPPβ, TNF-α, IL-6 and MCP-1, findings not observed in untreated cells. These data corroborate other studies reporting that unstimulated astrocytes did not produce cytokines unless treated with inflammatory mediators, such as those that we used [122]. In this condition, only miR-21 was found to be enriched in their derived exosomes and may further elicit a pro-inflammatory response [123].

Overall, it was remarkable that the three AD iPSC-derived cell types (microglia, neurons, and astrocytes) evidenced similar miR-21 overexpression. Other miRNAs additionally characterize each cell type. This is the case for upregulated miR-146a in microglia, elevated miR-124 and miR-125b in neurons and downregulated miR-155 in astrocytes. Notably, in contrast to these cell-type-specific miRNAs, miR-21 was the only one that stood out in organotypic slices transplanted with SWE cells and in CSF samples from MCI-AD patients. This study is thus pioneer in identifying miR-21 upregulation in several tested models of human AD, thus highlighting its potential relevance as a crucial player in AD pathophysiology. Further studies should explore whether this overexpression in cells is part of the defense mechanism against AD or if, in contrast, it is implicated in the loss of function and accumulation of harmful proteins. Also deserving to be explored is whether miR-21 inclusion in cell-derived exosomes and its dissemination to target cells is protective against neuroinflammation or further aggravates associated processes in these recipient cells. As we performed for neuronal miR-124 [25,40], mimics and inhibitors of miR-21 should be tested in human AD models, such as those that we used here, as well as in 3D human triculture systems modeling neurodegeneration and neuroinflammation in AD [124].

When we assessed whether the upregulation of miR-21 regulates the expression of PPARα, one of its specific targets [73], we observed its consistent downregulation in human and murine cells in mouse organotypic HCs after SWE cell xenotransplantation. Similar results were observed for MG-AD cells and stressed AST-AD, confirming the regulatory effects of miR-21 in these models. It is noted, however, that in non-stressed AST-AD, the elevated levels of miR-21 did not translate into a PPARα reduction. This may derive from the combined influence of the miR-155 reduction, which can concur to elevate PPARα [125], thus annulling the effect of miR-21 and causing the elevation of the PPARα gene in naïve AST-AD. This contrasts with the marked decrease in stressed astrocytes, which no longer revealed the miR-155 reduction. The suppression of PPARα in astrocytes was suggested to impair the astroglial uptake and degradation of Aβ [75], as well as to inhibit the regulation of microglial inflammatory properties by PPARα ligands, which negatively regulates nuclear factor-kappa B and activator protein-1 pathways [126,127], thus contributing to AD onset and progression. Indeed, decreased expression of PPARα was found in the brains of AD patients and was associated with weaker anti-oxidative and anti-inflammatory processes, as well as mitochondrial dysfunction and dysregulated APP processing [128]. In NEU-AD cells, the only cells revealing elevated miR-124 and miR-125b together with miR-21, we noted that miR-21 did not significantly target the expression of PPARα. This may have been due to the simultaneous upregulation of miR-125b with known target sites in the PPARα gene [129].

With a consistent increase in every used model and in patient samples, our study highlights miR-21 as a promising miRNA to explore as a disease-modifying target in AD, given its implications in neuronal dysfunction, microglial activation and astrocyte reactivity, as well as its inclusion in cell-derived exosomes, thus enabling its propagation. miR-21-based therapeutic approaches have already demonstrated success in different disease fields [130,131,132] and initiated at least two clinical trials, one in Alport Syndrome (NCT03373786) and another in Diabetic Wound Healing (NCT02581098). Frequently, miR-21 is a matter of debate due to its lack of specificity and simultaneous description of beneficial and harmful properties [33,66,67,133,134,135,136]. Similar attention should also be given to miR-21 in the AD field, and future studies should now aim at its modulation with further evaluation of the molecular, cellular, and cognitive consequences.

## 5. Conclusions

Two of the major problems in finding effective target-driven therapies in AD are the absence of models recapitulating most disease hallmarks and the difficulties working with human brain samples. Treatments that effectively slow neurodegeneration and cognitive impairment in AD are lacking. Combined cell-based human models may be pivotal in identifying new potential biomarkers and developing drug discovery strategies. Recent evidence highlights that miRNA impairment may be closely related to AD pathogenesis, but strong differences between healthy and AD patient miRNA profiles are still lacking.

Here, we used different promising translational models, from hippocampal slices cultured with human SWE cells to CSF from MCI-AD patients, as well as MG-AD, Neu-AD and AST-AD originating from iPSCs of *PSEN1ΔE9* AD patients. Our aim was to identify responsive cell biomarkers that could be correlated to cell-specific and translational miRNAs, with an emphasis on miR-21 and miR-124, according to our previously published data [25,39,40]. Our study highlights the transversal relevance of miR-21 among our selected set of miRNA hits and validates its potential as a target/biomarker in AD pathophysiology. With WT mouse organotypic cultures transplanted with SWE cells, we attempted to recapitulate the early events leading to neuropathological manifestations caused by AD neuroblastoma cells. Glial activation and upregulated miR-21 were observed as the first outcomes. Since the dysfunction of miRNAs has been increasingly recognized in AD as diagnostic markers or therapeutic agents, we then confirmed the discriminatory potential of CSF miR-21 by observing marked differences in miR-21 between MCI-AD and non-AD MCI individuals. A question crosses our minds: Which cell/cells could contribute to this? By using MG-AD, Neu-AD and Ast-AD cells generated from the same AD patients, we were able to validate the contribution of all by identifying the transversal upregulation of miR-21. Its regulatory effects on its PPARα target were manifested in MG-AD and AST-AD cells after immunostimulation with C1q + IL-1α + TNF-α. These stressed astrocytes showed an increased number of fibroblast-like cells and of S100B-positive cells, together with dysregulated gene expression and the release of sAPPβ and cytokines, indicating their reactive and neurotoxic phenotype. The lack of PPARα changes in Neu-AD may derive from the concomitant upregulation of miR-125b and, in the non-stressed AST-AD, from the coexistent miR-155 downregulation condition.

Signaling by miR-21 overexpression may extend from the cells to the microenvironment and to target cells through its travel within exosomes. This was verified in those from Neu-AD and from immunostimulated Ast-AD cells, thus supporting miR-21’s emerging role in mediating regulatory/pathological functions in such AD cells, but also in promoting similar effects on near and distant recipient cells.

In summary, despite the criticism involving miR-21 as a biomarker for most diseases, here, we provide a multi-model dataset confirming its potential as a therapeutic target to explore in AD. To the best of our knowledge, this is the first study confirming miR-21 elevation in AD using so many different research platforms. Considering the broad expression of miR-21, we support a personalized and strict CNS miR-21-based strategy aimed at regulating its expression toward homeostatic levels, thus limiting possible off-target consequences. In the future, the testing of exosomes loaded with miR-21 mimics and inhibitors in 2D/3D triculture microfluidic devices and administration in AD mouse models, such as 5xFAD mice, will elucidate the overall role of miR-21 in integrated cell systems, either in reprogramming dysregulated microglia/astrocyte activation or in delaying the hallmarks of AD progression and neurocognitive outcomes.

## Figures and Tables

**Figure 1 cells-11-03377-f001:**
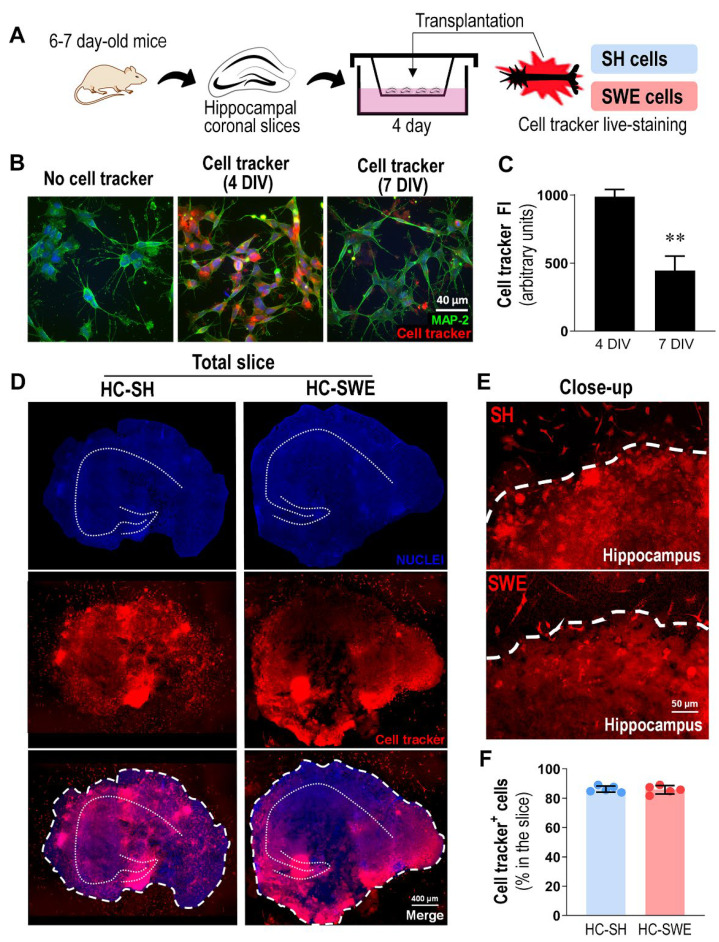
SH and SWE neuroblastoma cells were efficiently transplanted into mouse organotypic hippocampal coronal slices. (**A**) Schematic representation of the transplantation process after SH/SWE live-cell staining with cell tracker. (**B**) Representative fluorescence images showing cell tracker (red) after 4 and 7 days in vitro (DIV). Neuroblastoma cells were immunostained with MAP-2, and nuclei were counterstained with Hoechst. (**C**) Cell tracker fluorescence quantification in SH/SWE after 4 and 7 DIV. (**D**) Representative fluorescence images of HC-SH or HC-SWE. Dotted lines represent a schematic outline of the hippocampal structure, and bold dashed lines delimit the margins of the organotypic hippocampal slices. Separate images are provided for nuclei and cell tracker, complemented with a merged image. (**E**) Representative close-up images for HC-SH and HC-SWE showing cell-tracker-positive SH or SWE cells inside and outside the margins of each hippocampal slice, with (**F**) respective quantification of cells grafted in the hippocampal slice. Scale bars correspond to 40 µm (**B**), 400 µm (**D**) and 50 µm (**E**). Results are mean ± SEM from five independent experiments. Two-tailed Student’s *t*-test: ** *p* < 0.01 vs. 4 DIV. SH, SH-SY5Y neuroblastoma cells; SWE, SH-SY5Y cells expressing human *APP695 Swedish* mutation; DIV, days in vitro; FI, fluorescence intensity; HC-SH, hippocampal cultures with engrafted SH cells; HC-SWE, hippocampal cultures with engrafted SWE cells; MAP-2, microtubule-associated protein 2.

**Figure 2 cells-11-03377-f002:**
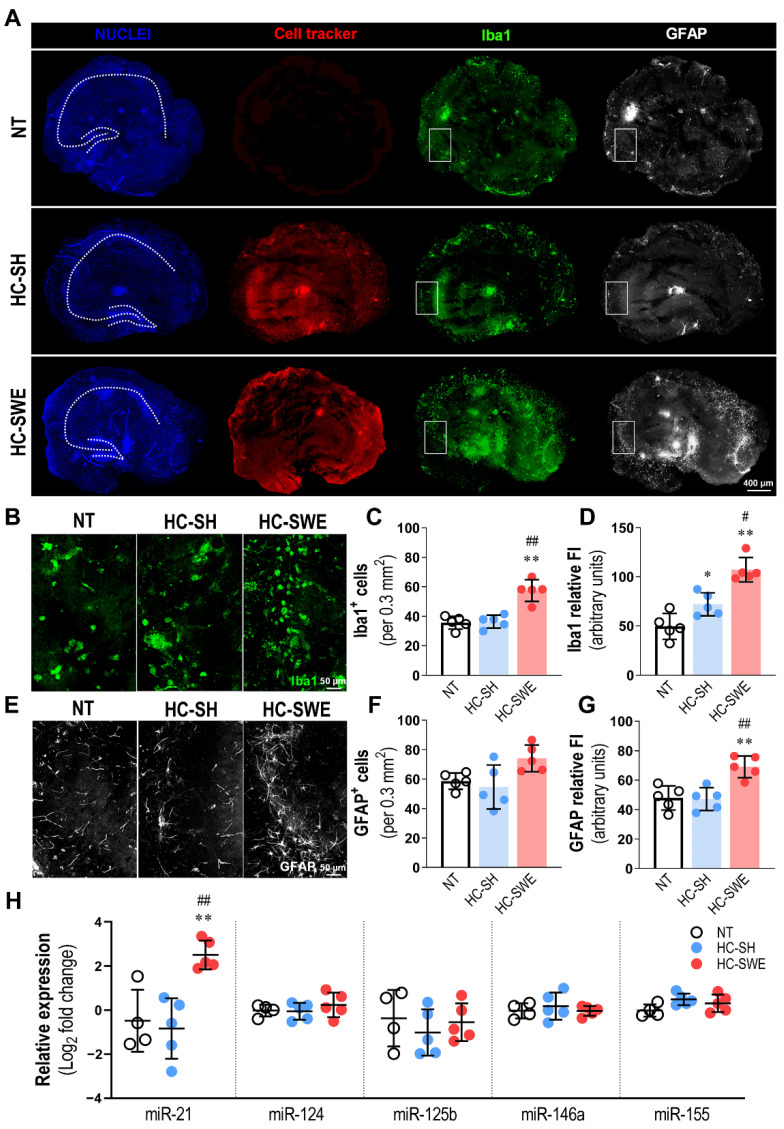
HC-SWE slices evidence an increased number of Iba1+ cells, enhanced Iba1/GFAP fluorescence intensities and upregulated miR-21 expression as compared with NT and HC-SH slices. (**A**) Representative fluorescence images of non-transplanted (NT) and hippocampal cultures with engrafted cells (HC-SH and HC-SWE). Nuclei (in blue), cell tracker+ (in red), Iba1+ (in green) and GFAP+ (in white), all shown separately. Dotted lines represent a schematic outline of the hippocampal structure. White rectangles are sections from similar hippocampal regions used for the close-up visualization. (**B**) Close-up images of Iba1+ cells. (**C**) Total quantification of the number of Iba1+ cells and (**D**) total Iba1 fluorescence intensity. (**E**) Close-up images of GFAP+ cells, (**F**) total quantification of the number of GFAP+ cells and (**G**) total GFAP fluorescence intensity. (**H**) miRNA quantification in HC-SWE vs. NT hippocampi, presented as the binary logarithm of fold change. Scale bars correspond to 400 µm (**A**) and 50 µm (**B**,**E**). Results are mean ± SEM from five independent experiments. Two-tailed Student’s *t*-test: * *p* < 0.05 and ** *p* < 0.01 vs. NT; # *p* < 0.05 and ## *p* < 0.01 vs. HC-SH. SH, SH-SY5Y neuroblastoma cells; SWE, SH-SY5Y cells expressing human *APP695 Swedish* mutation; NT, non-transplanted hippocampal slices; HC-SH, hippocampal slices transplanted with SH cells; HC-SWE, hippocampal slices transplanted with SWE cells; FI, fluorescence intensity; GFAP, Glial fibrillary acidic protein; Iba1, ionized calcium binding adaptor molecule 1.

**Figure 3 cells-11-03377-f003:**
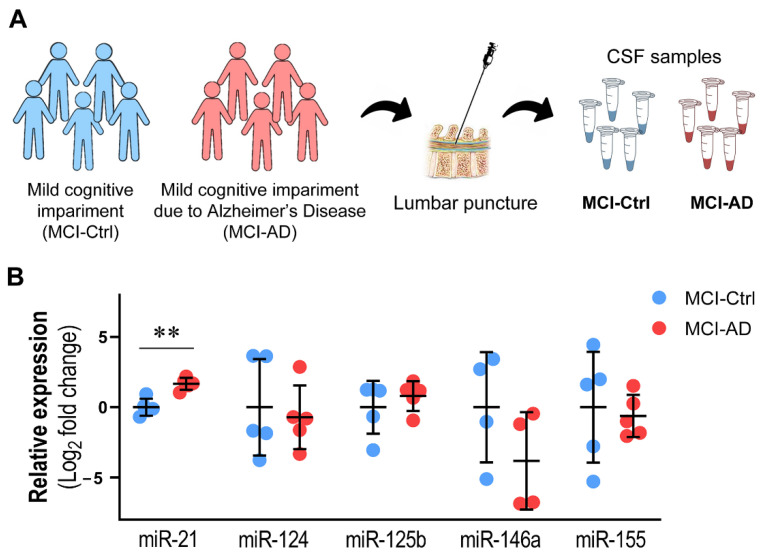
Exploratory miRNA profile in the cerebrospinal fluid (CSF) from MCI patients who fulfilled criteria for MCI due to AD (MCI-AD) vs. MCI patients with no biomarker criteria for AD (MCI-Ctrl). (**A**) Schematic representation of the MCI-AD and MCI-Ctrl series and the procedure to obtain CSF samples by lumbar puncture. (**B**) The miRNA profile in the CSF samples from MCI-AD (red dots) side-by-side with the miRNA profile in the CSF from MCI-Ctrl (blue dots). The results are mean ± SEM presented as the binary logarithm of fold change from five different subjects of each group. Mann–Whitney U test with Bonferroni post hoc correction: ** *p* < 0.01, MCI-AD vs. MCI-Ctrl. AD, Alzheimer’s disease; Ctrl, control; MCI, mild cognitive impairment.

**Figure 4 cells-11-03377-f004:**
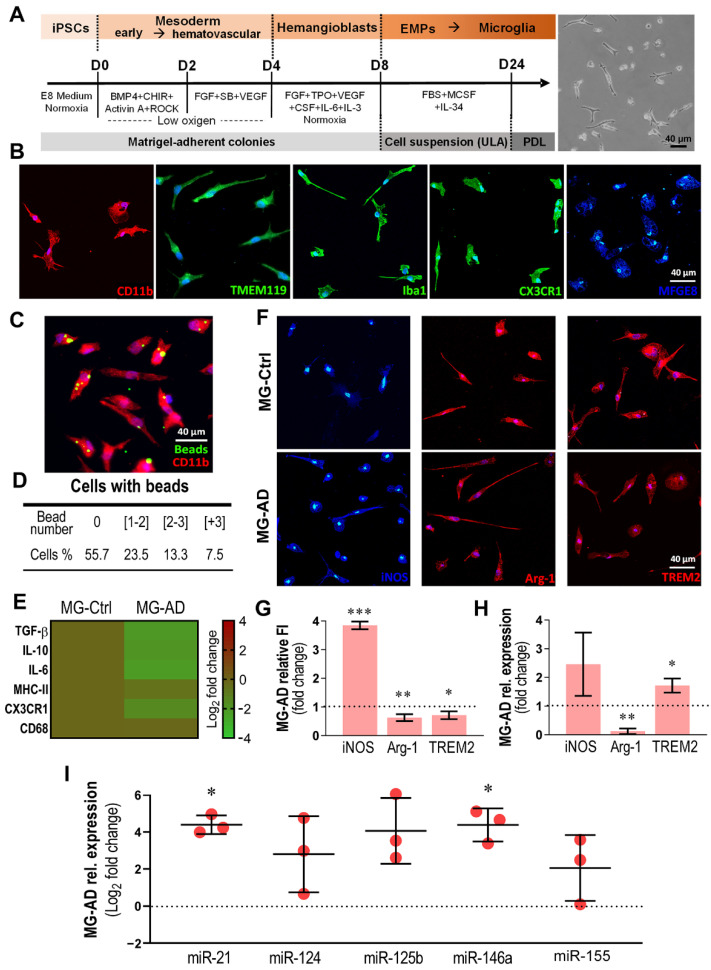
MG-AD cells show increased iNOS, miR-21 and miR-146a, as well as decreased Arg-1 and TREM2, consistent with an inflammatory signature. (**A**) Schematic representation of iPSC differentiation into microglia (MG), as detailed in Materials and Methods. (**B**) Characterization of microglial cells for CD11b, TMEM119, Iba1, CX3CR1 and MFGE8 markers. Nuclei were counterstained with Hoechst. (**C**) Representative image of microglia phagocytosing green zymosan fluorescent beads. (**D**) Percentage of cells phagocytosing a distinct number of beads. (**E**) Inflammatory gene heatmap representation (TGF-β, IL-10, IL-6, MHC-II, CX3CR1 and CD68) in MG-AD vs. MG-Ctrl obtained by RT-qPCR. (**F**) Representative images of iNOS, Arg-1 and TREM2 immunofluorescence signal in AD-MG and Ctrl-MG and (**G**) their respective quantification. (**H**) Gene expression data for iNOS, Arg-1 and TREM2 for MG-AD vs. MG-Ctrl (dotted line), determined by RT-qPCR. (**I**) Quantification of immune-related miRNAs in AD-MG vs. Ctrl-MG (dotted line), shown as the binary logarithm of fold change. All scale bars correspond to 40 µm. Results are mean ± SEM from three independent experiments performed on iPSC-derived microglia from an AD female patient carrying the *PSEN1ΔE9* deletion and in a matched control. Two-tailed Student’s *t*-test: * *p* < 0.05, ** *p* < 0.01 and *** *p* < 0.001, MG-AD vs. MG-Ctrl (dotted line). MG, iPSC-induced microglia; AD, Alzheimer’s disease; Ctrl, control; BMP4, bone morphogenetic protein 4; CHIR, glycogen synthase kinase 3β inhibitor; ROCK, rho-associated protein kinase inhibitor; FGF, fibroblast growth factor; SB, SB431542; VEGF, vascular endothelial growth factor TPO, thrombopoietin; CSF, colony-stimulating factor; FBS, fetal bovine serum; MCSF, macrophage-colony-stimulating factor; EMPs, erythro-myeloid progenitors; ULA, ultra-low-attachment plates; PDL, poly-D-lysine; TGF-β, transforming growth factor β; IL, interleukin; MHC-II, major histocompatibility complex class II; CX3CR1, CX3C motif chemokine receptor 1; CD68, cluster of differentiation 68; iNOS, inducible nitric oxide synthase; TREM-2, triggering receptor expressed on myeloid cells 2; Arg-1, arginase; MFGE8, milk fat globule-EGF factor 8 protein; Iba1, ionized calcium binding adaptor molecule 1.

**Figure 5 cells-11-03377-f005:**
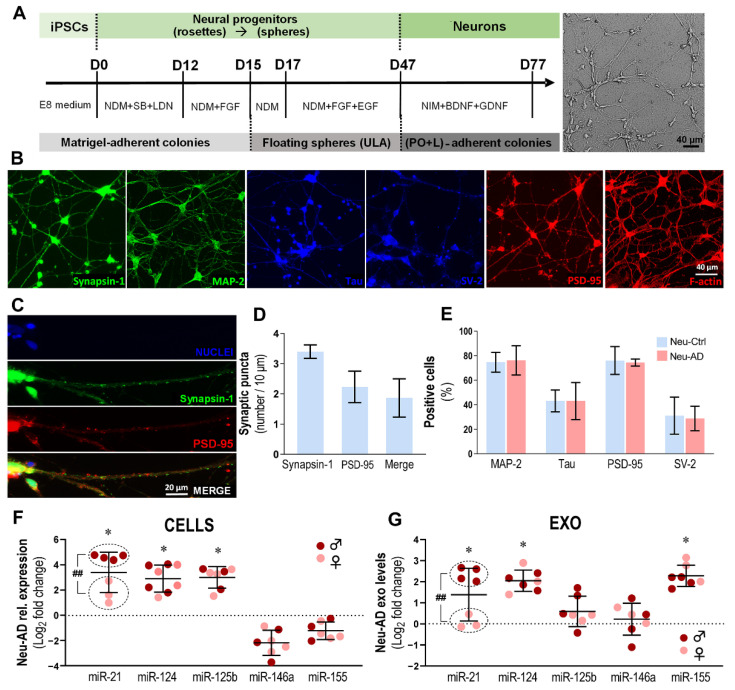
Characterization reveals successfully differentiated and maturated neurons, while Neu-AD cell lines show alterations in intracellular and exosomal miRNAs, which include the simultaneous elevation of miR-21 and miR-124. (**A**) Schematic representation of iPSC differentiation into induced neurons (Neu), as detailed in Materials and Methods. (**B**) iPSC-derived neurons express the typical neuronal markers synapsin-1, MAP-2, Tau, SV-2, PSD-95 and F-actin. (**C**) Representative images of synaptic puncta in neurons, with co-localization of synapsin-1 and PSD-95, and (**D**) quantification of individual and double-labeled puncta. Nuclei were counterstained with Hoechst. (**E**) Comparative number of cells expressing MAP-2, Tau, PSD-95 and SV-2 protein markers in neurons from AD and Ctrl donors. (**F**) Quantification of the selected inflamma-miRNAs in Neu-AD cell lines and (**G**) in their derived exosomes expressed as the binary logarithm of fold change vs. Neu-Ctrl cells (dotted line). Samples from a male patient (in blue dots) and a female patient (in pink dots) were distinguished. Scale bars correspond to 40 µm in (**A**,**B**) and to 20 µm in (**C**). Results are mean ± SEM from at least 7 independent experiments. Two-tailed Student’s *t*-test: * *p* < 0.05, Neu-AD vs. Neu-Ctrl; ## *p* < 0.01, male (n = 4) vs. female (n = 3) samples. Neu, iPSC-induced neurons; AD, Alzheimer’s disease; Ctrl, control; EXO, exosomes. SV-2, synaptic vesicle protein 2; PSD-95, postsynaptic density protein; MAP-2, microtubule-associated protein-2; F-actin, filamentous actin; NDM, neural differentiation media; LDN, LDN193189 dihydrochloride; SB, SB431542; FGF, fibroblast growth factor; EGF, epidermal growth factor; NIM, neural induction media; BDNF, brain-derived neurotrophic factor; GDNF, glial-cell-derived neurotrophic factor; ULA, ultra-low-attachment plates; PO, poly-L-ornithine; L, laminin; D, day.

**Figure 6 cells-11-03377-f006:**
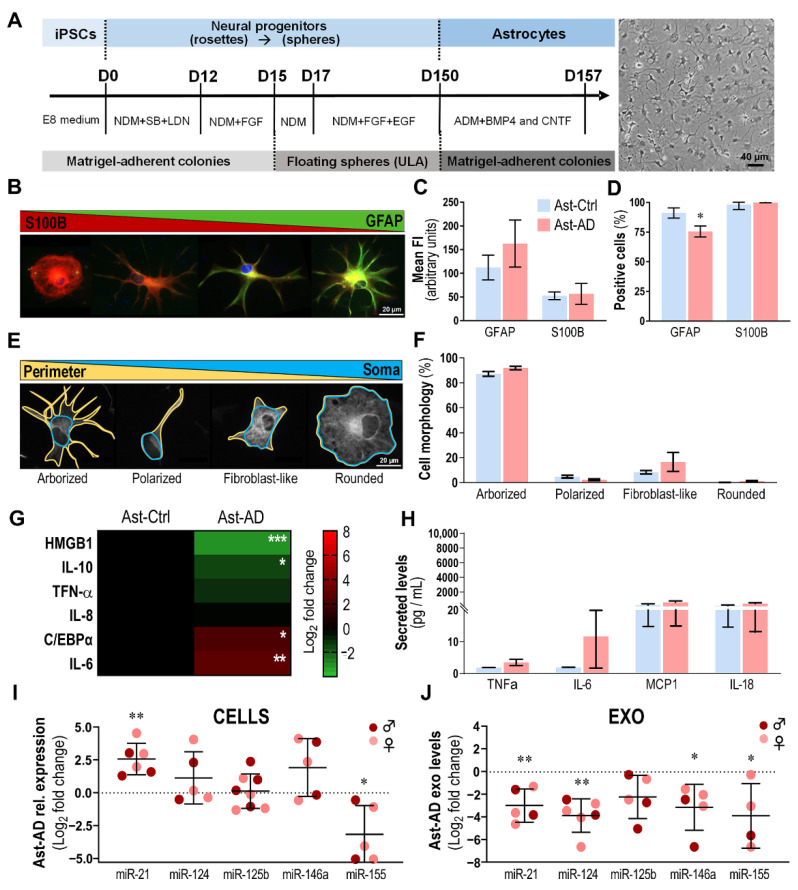
Ast-AD cells show fewer GFAP-positive cells and dysregulated immune-related markers, together with dissimilar miR-21 profiles in cells and exosomes. (**A**) Schematic representation of iPSC differentiation into induced astrocytes (Ast), as detailed in Materials and Methods. (**B**) Astrocytes display a range from almost exclusively S100B+ cells to GFAP+ cells. Nuclei were counterstained with Hoechst. (**C**) GFAP and S100B mean fluorescent intensities (FIs) of Ast-Ctrl and Ast-AD cells. (**D**) Number of GFAP- and S100B-positive cells in Ast-Ctrl and Ast-AD cells. (**E**) Morphological diversity of astrocytes based on the cell perimeter (yellow outline) and cell soma (blue outline). Cells with more than two branches were considered arborized; cells with a single branch and an acentric nucleus were classified as polarized; cells with no ramifications and an irregular shape were considered fibroblast-like; and cells with a circular shape were classified as rounded. (**F**) Percentage of arborized, polarized, fibroblast-like and rounded AD and Ctrl astrocytes. (**G**) Heatmap representation of differentially expressed immune-related genes in Ast-AD vs. Ast-Ctrl cells. (**H**) Inflammatory-associated cytokine profile in the secretome of Ast-AD cells vs. that of Ast-Ctrl cells, determined by LEGENDplex flow cytometry assay. (**I**) Cell quantification of inflamma-miRNAs in Ast-AD vs. Ast-Ctrl (dotted line) in cells and (**J**) in exosomes, expressed as the binary logarithm of the obtained fold change. No differences in AD samples were found between male/female miRNA data. Scale bars correspond to 40 µm (**A**) and 20 µm (**B**,**C**). Results are mean ± SEM from at least five independent experiments. Two-tailed Student’s *t*-test: * *p* < 0.05, ** *p* < 0.01 and *** *p* < 0.001, Ast-AD vs. Ast-Ctrl, for all. iPSC, induced pluripotent stem cells; D, days; Ast, iPSC-induced astrocytes; AD, Alzheimer’s disease; Ctrl, control; EXO, exosomes; NDM, neural differentiation media; SB, SB431542; LDN, LDN193189 dihydrochloride; FGF, fibroblast growth factor; EGF, epidermal growth factor; ADM, astrocyte differentiation medium; BMP4, bone morphogenetic protein 4; CNTF, ciliary neurotrophic factor; ULA, ultra-low-attachment plate; HMGB1, high-mobility group box 1; IL, interleukin; TNF-α, tumor necrosis factor alpha; C/EBPα, CCAAT/enhancer-binding protein alpha; MCP-1, monocyte chemoattractant protein-1.

**Figure 7 cells-11-03377-f007:**
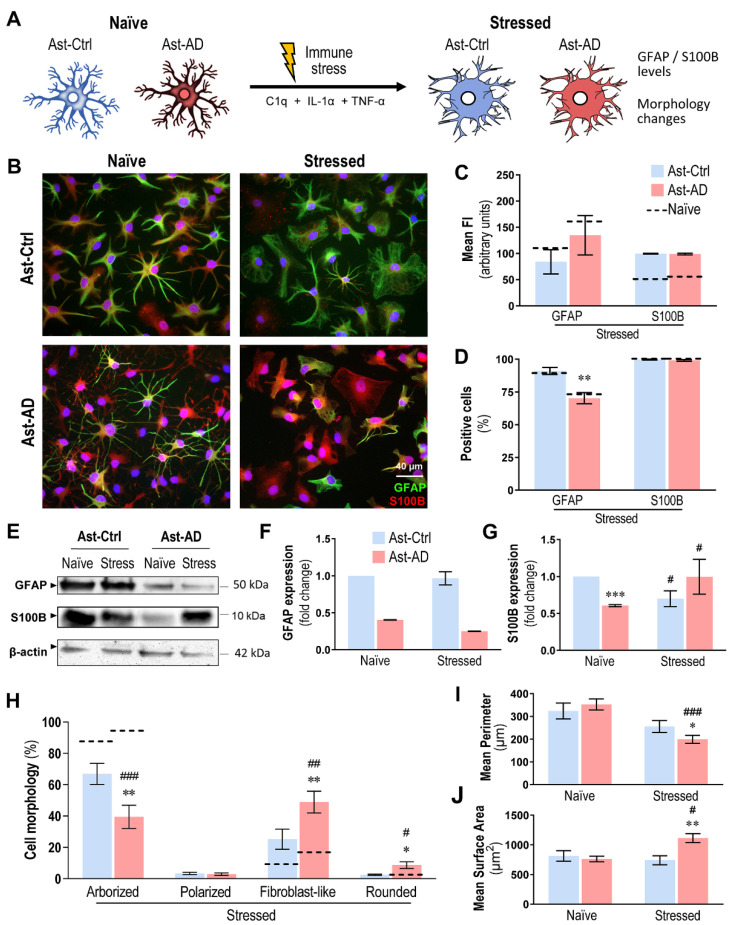
Immunostimulation with C1q + IL-1α + TNF-α maintains a reduced number of GFAP-positive Ast-AD cells, while it enhances S100B expression and switches the arborized Ast-AD stressed cells toward fibroblast-like and round cells. (**A**) Schematic representation of the 48 h immunostimulation model of astrocytes from AD and Ctrl samples (Ast-AD and Ast-Ctrl, respectively), as detailed in Materials and Methods. (**B**) Representative immunofluorescence images of naïve (non-immunostimulated) and stressed (immunostimulated) Ast-Ctrl and Ast-AD cells. Astrocytes were double-labeled with GFAP (green) and S100B (red). Nuclei were counterstained with Hoechst. (**C**) GFAP and S100B mean fluorescent intensities (FIs) of stressed Ast-Ctrl and Ast-AD cells vs. naïve ones (dashed line). (**D**) Number of GFAP- and S100B-positive cells in stressed Ast-Ctrl and Ast-AD cells vs. naïve ones (dashed line). (**E**) Representative Western blot images of GFAP and S100B cellular levels, with the respective (**F**) GFAP and (**G**) S100B densitometric quantifications. (**H**) Percentage of arborized, polarized, fibroblast-like and rounded cell morphologies after immunostimulation of Ast-Ctrl and Ast-AD cells vs. respective naïve cells (dashed lines). (**I**) Mean perimeter and (**J**) mean surface area of naïve and stressed Ast-Ctrl and Ast-AD cells. Scale bar corresponds to 40 µm (**B**). Results are mean ± SEM from at least 4 independent experiments. For GFAP Western blot quantification, a pool of 4 samples was used. Two-tailed Student’s *t*-test: * *p* < 0.05, ** *p* < 0.01 and *** *p* < 0.001, Ast-AD vs. Ast-Ctrl in same conditions; # *p* < 0.05, ## *p* < 0.01 and ### *p* < 0.001 immunostimulated vs. respective naïve cells; Ast, iPSC-induced astrocytes; Ctrl, control; AD, Alzheimer’s disease; C1q, complement component 1q; IL-1α; interleukin-1 alpha; TNF-α, tumor necrosis factor alpha; GFAP, glial fibrillary acidic protein; S100B, S100 calcium binding protein B.

**Figure 8 cells-11-03377-f008:**
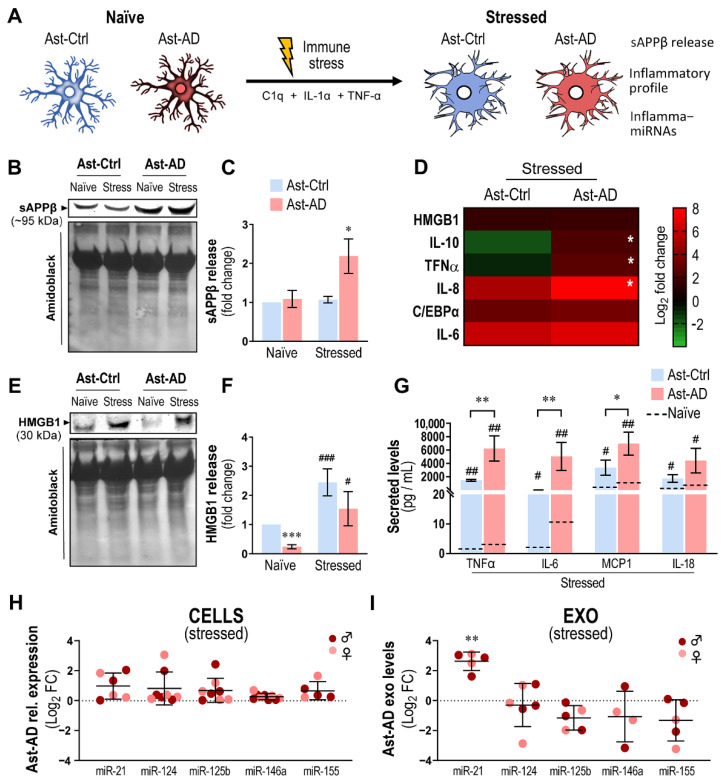
Ast-AD cells show increased inflammatory gene expression, sAPPβ release, cytokine secretion and exosomal enrichment in miR-21 after immunostimulation. (**A**) Schematic representation of the immune activation of both Ast-Ctrl and Ast-AD cells for 48 h with C1q + IL-1α + TNF-α, as detailed in Materials and Methods. (**B**) Representative Western blot images of sAPPβ released into the cell media, with the respective (**C**) densitometric quantification. Results were normalized to total secreted protein detected by Amido Black. (**D**) Heatmap representation of the differentially expressed inflammatory genes in Ast-AD vs. Ast-Ctrl after immune stress. (**E**) Representative Western blot images of HMGB1 released into the cell media, with the respective (**F**) densitometric quantification. (**G**) Secreted inflammatory-associated cytokines by stressed Ast-AD and Ast-Ctrl cells vs. respective naïve cells (dashed line), determined by the LEGENDplex flow cytometry assay. Quantification of inflamma-miRNAs in (**H**) stressed Ast-AD vs. Ast-Ctrl cells (dotted line), as well as in their (**I**) exosomes, expressed as the binary logarithm of fold change (FC). Results are mean ± SEM from at least 4 independent experiments. Two-tailed Student’s *t*-test: * *p* < 0.05, ** *p* < 0.01 and *** *p* < 0.001, Ast-AD vs. Ast-Ctrl; # *p* < 0.05, ## *p* < 0.01 and ### *p* < 0.001, immunostimulated vs. respective naïve cells; no differences were found between male and female samples. Ast, iPSC-induced astrocytes; Ctrl, control; AD, Alzheimer’s disease; EXO, exosomes. C1q, complement component 1q; IL-1α; interleukin-1 alpha; TNF-α, tumor necrosis factor alpha; sAPPβ, soluble amyloid precursor protein beta; HMGB1, high-mobility group box 1; IL, interleukin; C/EBPα, CCAAT/enhancer-binding protein alpha.

**Figure 9 cells-11-03377-f009:**
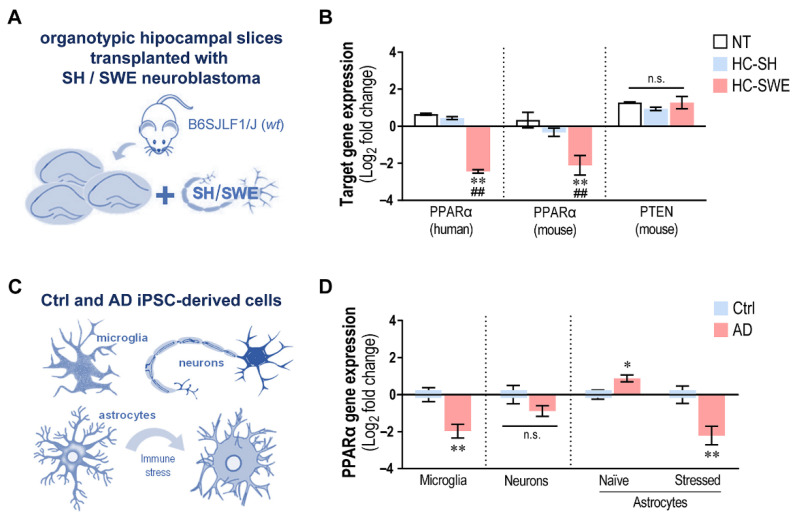
Gene expression of the miR-21 target PPARα is repressed in SWE-transplanted organotypic hippocampal cultures, in AD iPSC-derived microglia and in stressed iPSC-derived AD astrocytes. (**A**) Schematic representation of hippocampal xenotransplantation with the (**B**) respective gene expression of miR-21 targets: human PPARα, mouse PPARα and mouse PTEN in NT, HC-SH and HC-SWE slices. (**C**) Schematic representation of Ctrl and AD iPSC-derived microglia, neurons and astrocytes. Astrocytes were also immunostimulated with C1q + IL-1α + TNF-α cocktail, as detailed in Materials and Methods. (**D**) Gene expression of PPARα in iPSC-derived microglia, neurons and astrocytes (naïve and stressed) from AD patients vs. respective control (Ctrl) cells. Results are mean ± SEM from at least 4 independent experiments, expressed as the binary logarithm of fold change. Two-tailed Student’s *t*-test: * *p* < 0.05 and ** *p* < 0.01 vs. NT/Ctrl; ## *p* < 0.01 vs. HC-SH. (**B**) or respective Ctrl cells (**D**). SH, SH-SY5Y neuroblastoma cells; SWE, SH-SY5Y neuroblastoma cells expressing the human *APP695 Swedish* mutation; NT, non-transplanted hippocampal slices; HC-SH, hippocampal cultures with engrafted SH cells; HC-SWE, hippocampal cultures with engrafted SWE cells; AD, Alzheimer’s disease; Ctrl, control; PPARα, peroxisome proliferator-activated receptor alpha; PTEN, phosphatase and tensin homolog.

**Table 1 cells-11-03377-t001:** Demographic data and clinical assessment of mild cognitive impairment (MCI) patients.

Name	Patient	Sex	Age	MMSE Score	CDR Scale	PiB-PET	CSF tTau (pg/mL)	CSF pTau (pg/mL)	CSF Aβ42 (pg/mL)
MCI-Ctrl	LIS-105	F	65 y	25	0.5	−	214	37	1070
	LIS-106	F	78 y	*ND*	0.5	−	280	46	1190
	LIS-107	F	73 y	28	0.5	−	296	46	1010
	LIS-113	M	57 y	27	0.5	−	194	29	617
	LIS-115	M	60 y	22	0.5	−	126	19	696
MCI-AD	LIS-096	M	58 y	28	0.5	+	1100	140	450
	LIS-097	F	74 y	27	0.5	+	566	75	555
	LIS-102	F	71 y	27	0.5	+	1140	111	274
	LIS-103	M	72 y	*ND*	0.5	+	539	59	494
	LIS-104	M	72 y	*ND*	0.5	+	186	30	463

MCI-Ctrl, mild cognitive impairment control patients; MCI-AD, patients with mild cognitive impairment due to AD; MMSE, mini-mental state examination; CDR, Clinical Dementia Rating Scale; ND, not determined; F, female; M, male; y, years; PiB-PET, Pittsburgh Compound B positron emission tomography; CSF, cerebrospinal fluid; Tau, total Tau protein; pTau, phosphorylated Tau; Aβ42, amyloid-β42 peptide.

**Table 2 cells-11-03377-t002:** Summary of the health controls and Alzheimer’s disease (AD) patients used in the present study.

Patient	Suffix	Diff.	Sex	Age at Biopsy	Mutation Genotype	APOE Genotype	Health Status	Sample Origin	Reprogr. Method	Karyotype	Ref.
Ctrl1	Ctrl	Neuron Astrocyte	F	Adult	-	ε3/ε3	Healthy	Skin biopsy	SeV 1.0	46XX Normal	[27]
Ctrl3	Ctrl	Microglia	F	44 y	-	ε3/ε3	Healthy	Skin biopsy	SeV 1.0	46XX Normal	[26]
AD2	AD	Neuron Astrocyte	M	48 y	*PSEN1ΔE9*	ε3/ε3	EOAD	Skin biopsy	SeV 2.0	46XY Normal	[27]
AD3	AD	Microglia Neuron Astrocyte	F	47 y	*PSEN1ΔE9*	ε3/ε3	Pre-symptomatic AD	Skin biopsy	SeV 2.0	46XX Normal	[27]

Diff. cell type differentiation; APOE, apolipoprotein E; F, female; M, male; Y, years; SeV, Sendai virus.

## Data Availability

The authors declare that the data and materials supporting the results in the manuscript are available upon request.

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
