# Peer review of "Emerging Role of miR-21-5p in Neuron–Glia Dysregulation and Exosome Transfer Using Multiple Models of Alzheimer’s Disease"

_cells, 2022, doi:10.3390/cells11213377_

Round 1

Reviewer 1 Report

The manuscript “Emerging role of miR‐21‐5p in neuron‐glia dysregulation and exosome transfer using multi‐models of Alzheimer’s disease” by Garcia et al., investigated how miRNA miR-21 could be a promising biomarker using AD cell models with organotypic hippocampal culture to see multiple AD pathogenic features. In addition, they evaluated miR-21 up-regulation in some control and patient CSF samples. They also found the same phenotypes in the iPS experiments. Overall experimental design and results were clear and provided essential data. I don’t have any further comments. Thank you.

Author Response

We thank the reviewer for his/her positive appreciation.

Reviewer 2 Report

Garcia et al in the current study (Emerging role of miR‐21‐5p in neuron‐glia dysregulation and 2 exosome transfer using multi‐models of Alzheimer’s disease) have established the unique mouse organotypic hippocampal slices and transplanted SWE cells system as well as the CSF from MCI patients and observed the activated subsets of glial cells and their link to the overexpression of the miR‐21. Furthermore, C1q, IL‐1α, and TNF‐α treated in such brain cell systems have shown the direct impact of miR 21 in the upregulation of S100B, sAPPβ, and cytokine releases. Overall, this is a well-written paper that provides useful insight into the field of AD.

Major:

C1q, IL‐1α, and TNF‐α are completely different immune players, and each one of the indicated pro-inflammatory mediators uses distinctive signaling cascades to propagate their effector functions. The author needs to provide that data on mouse organotypic hippocampal slices transplanted separately with different cell types, (e.g., Astrocytes, microglial cells, and neurons) and their separate treatment with (1) C1q, (2) IL‐1α, (3) TNF‐α, and (4) combined treatment (C1q, IL‐1α, and TNF‐α)  exhibiting whether there are differences related to cell types and/or treatment that cause alteration in miR21, cellular activation, and pro-inflammatory cytokines production. Overall well-designed paper but the needs additional data.

Author Response

Please see the attached response.

Reviewer 3 Report

This is a mingle-mangle of disconnected data following several incoherent lines of investigation without tying them together. About one third of the paper is describing experimental set up, methodology, and characterizing and confirming cellular phenotypes. Another third refers to intracellular and exosome measurements of miRNAs and another third to characterizing inflammatory parameters. The studies are mostly descriptive and are done in naïve and immune-stressed cells comparing Controls with AD stringing together a series of disconnected experiments and lengthy data presentations. The experiments on analyzing cohorts, i.e., CSF, and iPSC-derived cells, and sex differences, are underpowered. Altogether, there is little advance to the miRNA/neurodegeneration field.

A key would be to somehow connect the findings and come up with a coherent story. For example, no effort was made to look at miRNA regulation of targets associated with the observed changes in cell morphology, gene/cytokine expression/release, exosome-mediated and interactive cell functions, etc. What are the implications of the xenotransplantation studies with regards to the data from subjects and iPSC-derived cells? How does miRNA localization in exosomes drive processes? What are the interactive properties of the different cell types analyzed? What is the emerging role of miR-21 in neuron-glia dysregulation and exosome transfer with regards to AD, as advertised in the title? All this and a lot more has not been analyzed.

Without functional experiments demonstrating clear interactions, most of the data interpretation is speculative. Accordingly, the discussion is a lengthy and incoherent essay on multiple aspects of this patched together study without providing a clear story. It is partly lacking professionality and has elements of a high school assay.

The authors are not precise in their language. The terminology of iNeurons, iAstrocytes, and iMG is misleading, as it indicates production of induced cells using direct reprogramming which was not done in these studies. Another example is in lines 644-647: “Our previous study using human neuron‐microglia cocultures revealed that increased miR‐21 in SWE neuronal cells was secreted in their exosomes, …”. Neurons were not analyzed in this studies and SH cells are not neurons, even when differentiated.

Fig. 2H: Is miR-21 expressed in the SWE cells or in the mouse hippocampal cells? Although the PCR is designed to amplify human miR-21, it should be demonstrated or clarified which cells express the microRNA.

Fig. 5: It would be important to also stain the neuronal cultures with non-neuronal markers, e.g., GFAP, to demonstrate the purity of the neuronal population.

Fig. 6D, and Fig. 7: Was the lower numbers of mature GFAP+ astrocytes derived from AD iPSC due to reduced proliferation or differentiation capacity?

Minor:

Some of the literature can be updated to include more recent papers, e.g., with regards to iPSC, AD pathology, modeling, etc.

Lane 147: Mouse brain coronal sections can’t be cut in 400 mm sizes.

Lines 563-564: It should read Figure 2B,C.

Author Response

Please attached response.
